# FLASHOMNI: A UNIFIED SPARSE ATTENTION ENGINE FOR DIFFUSION TRANSFORMERS

## ABSTRACT

Multi-Modal Diffusion Transformers (DiTs) demonstrate exceptional capabilities in visual synthesis, yet their deployment remains constrained by substantial computational demands. To alleviate this bottleneck, many sparsity-based acceleration methods have been proposed. However, their diverse sparsity patterns often require customized kernels for high-performance inference, limiting universality. We propose **FlashOmni**, a unified sparse attention engine compatible with arbitrary DiT architectures. FlashOmni introduces flexible *sparse symbols* to standardize the representation of a wide range of sparsity strategies, such as feature caching and block-sparse skipping. This unified abstraction enables the execution of diverse sparse computations within a single *attention kernel*. In addition, FlashOmni designs optimized *sparse GEMMs* for attention blocks, leveraging sparse symbols to eliminate redundant computations and further improve efficiency. Experiments demonstrate that FlashOmni delivers near-linear, closely matching the sparsity ratio speedup (1:1) in attention and GEMM-$Q$, and achieves $2.5\times$–$3.8\times$ acceleration in GEMM-$O$ ($87.5\%$ of the theoretical limit). Applied with a multi-granularity sparsity strategy, it enables the Hunyuan model (33K) to achieve an avg $1.5\times$ end-to-end acceleration without degrading visual quality. FlashOmni is available at `https://anonymous.4open.science/r/FlashOmni-B980/`.

## 1 INTRODUCTION

Diffusion Transformers (DiTs) (Peebles & Xie, 2023) have achieved remarkable progress in high-fidelity visual generation (Esser et al., 2024; Black-Forest-Labs, 2024; Kong et al., 2024; Li et al., 2024; Batifol et al., 2025) by leveraging attention. However, its high computational complexity constrains inference efficiency, a challenge that grows with model scale, especially for high-resolution image and long video generation scenarios. To address this, various acceleration approaches have been developed, with sparsity-based methods standing out as the most universally applied techniques for their rapid, training-free deployment.

There are two distinct categories of sparse acceleration methods based on the granularity of sparsity applied during attention computation. *(i) Feature caching* (Chen et al., 2024; Selvaraju et al., 2024; Zou et al., 2024; 2025; Liu et al., 2025b) primarily exploits the feature similarity between adjacent timesteps, focusing on reusing or predicting the whole computations of selected tokens across steps to reduce the per-step computational burden by caching the corresponding feature. *(ii) Block-sparse skipping* leverages the inherent sparsity of attention computation, where many softmax operation outputs approach zero (Deng et al., 2024), to skip unimportant block-tile computations along the reduction axis via various evaluation strategies. As attention patterns vary considerably across different tasks, multiple sparse attentions have emerged, such as dynamic sparse attention (Xi et al., 2025; Zhang et al., 2025b; Xia et al., 2025; Xu et al., 2025; Yang et al., 2025) and pattern-based sparse attention (Yuan et al., 2024; Zhang et al., 2025a), each providing a task-specific sparsity design.

Despite promising progress, sparse acceleration for DiTs still faces several limitations: *(i) Fragmented and Constrained Sparse Strategy Design.* Kernels designed for block-skipping strategies are tightly restricted to this specific fine-grained sparsity pattern. Conversely, feature-caching approaches lack efficient kernel implementations. Without a unified kernel, these two strategies cannot be jointly applied within a single layer, severely limiting the flexibility and composability of sparse strategy designs. *(ii) Lack of sparse kernel generality.* Existing attention kernels cannot simultaneously

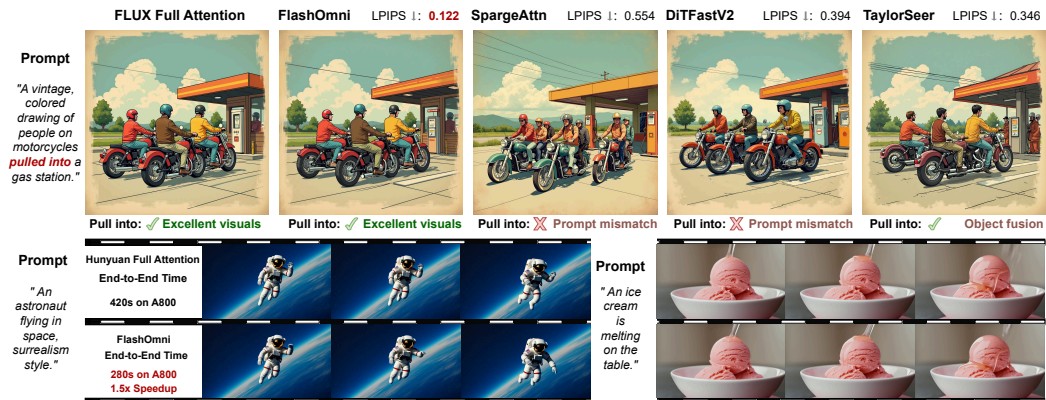

Figure 1: Visualization of different acceleration methods on FLUX and FlashOmni's speedup on HunyuanVideo.

support arbitrary sparse granularities and are not well optimized for the extra overhead (e.g., memory overhead). Moreover, without feature caching support, existing sparse kernel libraries lack optimization chance for sparse GEMM, failing to leverage coarse-grained sparsity to skip redundant computation in GEMM, much like KVCache in LLMs avoids recomputing for previous same tokens.

To address these challenges, we propose FlashOmni, a unified sparse attention engine for diffusion transformers, which supports the multi-step denoising using multi-granularity sparsity and introduces several key designs: *(i) Sparse symbols.* We introduce compact 8-bit sparse symbols to represent multiple levels of sparsity in a unified format. These symbols guide the selective update of cached features, enabling flexible multi-granularity integration. *(ii) Unified sparse attention kernel.* We design a single kernel that decodes sparse symbols at runtime and efficiently executes diverse sparsity strategies. It supports arbitrary sparsity patterns within one engine by treating feature caching as a new dimension of sparsity and integrate it into the same abstraction. *(iii) Optimized sparse GEMMs.* We further develop GEMM-*Q*/-*O*, which leverage sparse symbols to prune redundant computations along spatial and reduction dimensions similar to how KVCache in LLMs avoids recomputing, while also improving feature-cache storage logic. Our experiments demonstrate that our sparse kernel design achieves near-linear speedup with increasing sparsity, reaching a one-to-one match with theoretical computation reduction in GEMM-$Q$ and attention, up $2.5\times$ to $3.8\times$ acceleration in GEMM-$O$ (87.5% of theoretical speedup peak), and delivering $2\times$ attention speedup with nearly $1.5\times$ end-to-end gain in Hunyuan built upon FlashOmni.

Our contributions can be summarized as follows:

- We propose FlashOmni, a unified sparse attention engine for DiTs that achieves end-to-end high-quality generation alongside high-performance inference by supporting more flexible multiple granularities sparsity design within a single layer.

- We unify multi-granularity sparsity via sparse symbols and provide a unified attention kernel that can simultaneously support arbitrary sparsity granularities computation within a single kernel, while maintaining high-performance inference.

- We introduce sparse GEMM-*Q*/-*O* that exploit feature caching sparsity, unlocking untapped potential to eliminate redundant computations, a capability missing in existing sparse kernels.

## 2 RELATED WORKS

**Transformers in Diffusion Models.** Diffusion Transformers (Peebles & Xie, 2023), which follow scaling laws, have achieved notable success in image (Chen et al., 2024; Li et al., 2024) and video generation (Zheng et al., 2024; HaCohen et al., 2024). Recently, the Multimodal Diffusion Transformer (MMDiT), introduced by SD3 (Esser et al., 2024), offers greater advantages. It replaces the cross-attention module of DiTs by independently projecting visual inputs and text embeddings before concatenating them for self-attention. Prominent adopters include SD3 and Flux series (Black-Forest-Labs, 2024; Batifol et al., 2025) in image

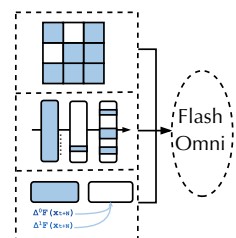

Figure 2: Typical sparse methods for DiTs.

synthesis, and CogVideoX (Yang et al., 2024), HunyuanVideo (Sun et al., 2024), and Mochi-1 (Team, 2024) in video generation. However, iterative sampling causes significant overhead in large and complex DiTs, limiting real-time applicability. Two primary sparse acceleration strategies as detailed below and illustrated in Figure 2.

**Feature Caching.** Feature caching leverages representation similarity between adjacent timesteps in DiTs and can be categorized into layer-wise and token-wise strategies. approaches (Ma et al., 2024; So et al., 2024; Li et al., 2023; Wimbauer et al., 2024) mitigate redundant computation via caching, interpolation, or low-frequency updates, with extensions that cache activations in both attention and MLP layers (Chen et al., 2024; Selvaraju et al., 2024). TeaCache (Liu et al., 2025a) further refines this by dynamically estimating timestep-dependent differences, while TaylorSeer (Liu et al., 2025b) enhances generation quality by forecasting future features from historical ones. In contrast, token-wise methods target the importance of individual tokens. ToCa (Zou et al., 2025) updates cached features based on attention-derived token scores, whereas Duca (Zou et al., 2024) approximates attention weights using the norm of the value matrix to estimate token importance.

**Block-Sparse Skipping.** Attention computation often exhibits inherent sparsity, with many softmax outputs approaching zero—a property widely exploited in ViTs (Beltagy et al., 2020; Hassani et al., 2023; Child et al., 2019) and LLMs (Zhang et al., 2023; Xiao et al., 2023; Fu et al., 2024). Sparse attention mechanisms for accelerating DiTs can be divided into static and dynamic approaches. Static methods predefine sparse patterns offline, such as prioritizing recent tokens. DiTFastAttn (Yuan et al., 2024) combines sliding-window patterns with attention sharing, while Sparse-vDiT (Chen et al., 2025) classifies attention heads into three predefined patterns. DiTFastAttnV2 (Zhang et al., 2025a) employs Arrow Attention with caching to enable head-wise selection in MMDiT, alleviating window constraints. Dynamic methods instead construct sparsity masks at runtime. Xattention (Xu et al., 2025) uses the sum of antidiagonal values in the attention matrix to provide a powerful proxy for block importance. SpargeAttention (Zhang et al., 2025b) derives masks from QK embeddings without relying on predefined patterns, whereas SVG2 (Yang et al., 2025) uses semantic-aware permutation to capture critical tokens better and reduce redundant computation.

## 3 METHOD

### 3.1 PRELIMINARY

Text-to-vision diffusion transformers (e.g., MMDiT) experience attention bottlenecks due to concatenating fixed-length text tokens ($N_{\text{text}}$) with resolution-dependent visual tokens ($N_{\text{vision}}$), producing a joint attention map with four regions: text-to-text, $v \to t$, $t \to v$, and vision-to-vision. Standard attention computes $S = QK^\top/\sqrt{d}$, $P = \text{Softmax}(S)$, $O = PV$, with $N = N_{\text{text}} + N_{\text{vision}}$. FlashAttention (Dao, 2024) accelerates this by block-partitioning $Q$, $K$, $V$ (sizes $b_q$, $b_k$) and applying online softmax (Milakov & Gimelshein, 2018). Acceleration strategies generally apply one sparsity pattern per block, ranging from coarse- to fine-grained:

**Definition 1** (Logical Block Sparse Masks). $M_c \in \{0,1\}^{\lceil N/b_q \rceil}$ marks cached output blocks; $M_s \in \{0,1\}^{\lceil N/b_q \rceil \times \lceil N/b_k \rceil}$ marks skipped $Q_i K_j^\top$ and $\widetilde{P}_{ij} V_j$ computations.

**Definition 2** (Sparse Strategies). Feature caching reuses $O_i$ when $M_c[i] = 0$, refreshing masks and cache at update steps. Block-sparse skipping omits pairs with $M_s[i,j] = 0$, with dynamic masks updated from the latest $Q$, $K$, and static masks pre-tuned offline for zero update cost.

### 3.2 FLASHOMNI OVERVIEW

Figure 3 presents the operational flow of FlashOmni, which adopts an *Update-Sparse* framework to integrate and execute two sparsity strategies within every single transformer layer:

*Update:* Given a sequence of adjacent timesteps $\{t, t-1, \ldots, t-\mathcal{N}\}$, FlashOmni first refreshes the sparse symbols and feature cache at the current step $t$. Using the newly computed $Q$ and $K$, a tailored sparsity-selection policy determines the sparsity type for each token block in the upcoming steps. Full attention is then applied to update the feature cache. When an output projection is present, FlashOmni additionally leverages GEMM-$O$ to further optimize cache updates.

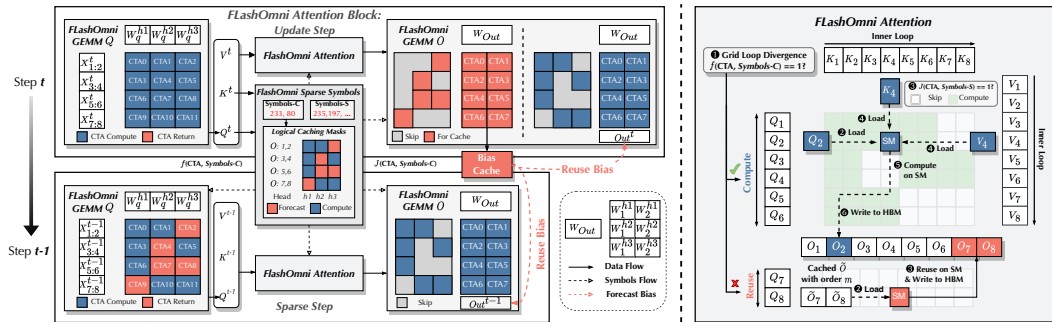

Figure 3: Detailed workflow of the FlashOmni framework: incorporating unified sparse symbols and sparse kernels (general sparse attention and GEMMs). Unified sparse symbols is refreshed only at *Update* timesteps, providing sparse guidance for corresponding sparse kernel executions at *Dispatch* timesteps.

*Sparse*: Utilizing the sparse symbols produced in the Update phase, FlashOmni accelerates attention computation over the following $\mathcal{N}$ timesteps $\{t-1, \ldots, t-\mathcal{N}\}$. Within the attention module, Cooperative Thread Arrays (CTA) adopt specialized computation modes, enabling different sparsity granularities for their respective block tiles. Guided by the caching symbols, FlashOmni also applies GEMM-$Q$ and GEMM-$O$ optimizations, removing redundant operations.

## 3.3 FLASHOMNI SPARSE SYMBOLS

FlashOmni employs two sparse symbols, $\mathcal{S}_c$ and $\mathcal{S}_s$, serving as a unified representation for multiple sparsity approaches, including feature caching and block-sparse skipping. Using MMDiT as a case study, we examine the joint attention map to determine the applicability of sparsity at two distinct granularities and derive the combination strategy adopted in FlashOmni.

Observation 1. In text-to-vision diffusion transformers, the $v \rightarrow t$ and $t \rightarrow v$ regions of the attention map are essential for reliable multimodal fusion. At each timestep, $v \rightarrow t$ updates text tokens via $P_{v \rightarrow t} V_{\text{vision}}$, embedding visual context into $O_{\text{text}}$. Conversely, $t \rightarrow v$ updates vision tokens via $P_{t \rightarrow v} V_{\text{text}}$, injecting textual guidance into $O_{\text{vision}}$. These two interactions are complementary: omitting $v \rightarrow t$ prevents text from perceiving visual changes, whereas omitting $t \rightarrow v$ leads to vision outputs misaligned with textual prompts. Empirically, we observe that caching image tokens that significantly influence text tokens, or are strongly affected by control-signal tokens, degrades cross-modal consistency. Therefore, FlashOmni excludes such tokens from caching to ensure timely and accurate multimodal updates.

*Logical Masks Generation.* FlashOmni first infers sparsity patterns from the attention structure and encodes them into logical block-sparse masks. These patterns are derived from a compressed attention map, which serves as the basis for constructing the sparse strategy combination used in this work. Specifically, every $n$ consecutive $\{Q_i\}$ and $\{K_j\}$ blocks (e.g., two blocks in Figure 3) are aggregated into single tokens $q$ and $k$ through token-gathering operations such as mean pooling, with pooling sizes $b_q$ and $b_k$. The compressed tokens $\{q_i\}$ and $\{k_j\}$ form a reduced attention map $\hat{P} = \text{Softmax}(\hat{S})$, where $\hat{S} = \{q_i\}\{k_j^T\} \in \mathbb{R}^{\lceil N/(nb_q) \rceil \times \lceil N/(nb_k) \rceil}$. From this representation, we define two metrics for guiding feature caching: (i) Vision-to-Text Contribution ($\mathcal{C}_{i,v \rightarrow t}$): for vision block $i$, this measures its contribution to text blocks based on the compressed attention map. Lower values indicate less influence on text tokens and are therefore prioritized for caching. Denoting $\alpha_{i,j}$ as the $(i,j)$ element in $\hat{P}[:\ n_t, n_t:]$, where $n_t = \lceil n_t/(nb_q) \rceil$ is the number of compressed text blocks, the metric is computed as $\mathcal{C}_{i,v \rightarrow t} = \sum_{j=1}^{n_t} \alpha_{j,i}$. (ii) Text-to-Vision Guidance ($\mathcal{G}_{i,t \rightarrow v}$): this reflects the extent to which vision block $i$ is influenced by text blocks. Blocks under stronger textual guidance retain attention computation to preserve responsiveness. With $\beta_{i,j}$ denoting the $(i,j)$ element in $\text{Softmax}(\hat{P}[n_t:,: n_t]^T)$, the score is given by $\mathcal{G}_{i,t \rightarrow v} = \sum_{j=1}^{n_t} \beta_{j,i}$. Given $\mathcal{C}_{i,t \rightarrow v}$ and $\mathcal{G}_{i,v \rightarrow t}$,

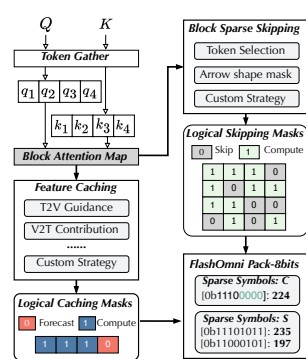

Figure 4: Example of FlashOmni sparse symbols generation for a single head of attention.

FlashOmni selects the indices with the lowest combined scores such that their cumulative sums do not exceed thresholds $\tau_c \cdot \sum_i \mathcal{C}_{i,t \to v}$ and $\tau_c \cdot \sum_i \mathcal{G}_{i,v \to t}$, as formalized in Equation 1:

$$\{ i \mid \mathrm{CumSum}_\uparrow(\mathcal{C}_{i,t \to v}) \leq \tau_c \cdot \sum_j \mathcal{C}_{j,t \to v} \ \wedge \ \mathrm{CumSum}_\uparrow(\mathcal{G}_{i,v \to t}) \leq \tau_c \cdot \sum_j \mathcal{G}_{j,v \to t} \} \quad (1)$$

Blocks with minimal scores are cached for use in the next step, while others undergo full attention computation. In the logical caching mask $M_c[i]$, cached positions are assigned 0 and non-cached positions 1. For cached blocks, FlashOmni forecasts future features via Taylor series expansion using cached features and their derivatives. For block-sparse skipping, token selection follows the compressed attention map.

*Sparse Symbols Compression.* To reduce storage overhead, the logical masks are encoded as 8-bit sparse symbols: $\mathcal{S}_c$ for feature caching and $\mathcal{S}_s$ for block-sparse skipping (Figure 4). In each update step, the latest $Q$ and $K$ are block-aggregated to form the attention map, from which masks are generated via the two-granularity sparsity strategy. For example, $M_c[3] = 0$ skips computation for $Q_7$ and $Q_8$, reusing $O_7$ and $O_8$ at the next sparse step. With big-end alignment, zero-padding yields the binary `0b11100000`, stored as the uint8 `224` for $\mathcal{S}_c$; $\mathcal{S}_s$ can similarly be encoded as `235` and `197`.

### 3.4 FLASHOMNI ATTENTION

FlashOmni implements a general attention kernel that can interpret sparse symbols and execute arbitrary sparse computations efficiently. In the FlashAttention framework, each CTA processes a complete computation tile. To support different computation granularities within a single kernel, we extend this design so that different CTAs can detect and adapt to the sparsity patterns associated with their respective granularities. At the same time, we treat feature caching as a new dimension of sparsity and integrate it into the same abstraction, so that CTAs can uniformly handle both block-sparse skipping and feature caching under a single symbolic interface. This unified design enables flexible, mixed-grained sparse computation, as outlined in Algorithm 1. Specifically, before loading $Q_i$, each CTA invokes the spatial-axis decoding function $\mathcal{F}$ (Line 5) and checks $\mathcal{S}_c$ to determine the sparse type of the current tile. Depending on this result, execution follows either a feature caching or a block-sparse skipping branch. The decoding operation uses only bitwise procedures, expressed as $\mathcal{F}(\mathcal{S}_c, i) = (\mathcal{S}_c >> i/n) \ \& \ 1$.

In the feature caching branch, element-wise computation logic is fused into the FlashOmni attention process, allowing each CTA to choose between standard attention or lightweight element-wise operations (e.g., summation and

---

**Algorithm 1:** FlashOmni Attention

**Input:**
1   $\{Q_i\} \in \mathbb{R}^{T_q \times b_q \times d}; T_q = N/b_q$
2   $\{K_i\}, \{V_i\} \in \mathbb{R}^{T_{kv} \times b_k \times d}; T_{kv} = N/b_k$
3   Sparse symbols: $\mathcal{S}_c, \mathcal{S}_s$; Cached output: $\widetilde{O}$;
**Output:** $O$
4 **for** $i \leftarrow 1$ **to** $T_Q$
5    **if** $\mathcal{F}(\mathcal{S}_c, i) == 0$ **then**
6      *# Feature Caching*
7      Continue if applying the GEMM-$O$;
8      Load $\widetilde{O}_i$ into the SM ;
9      $O_i = \mathrm{OP}_{\text{reuse}}(\widetilde{O}_i)$;
10   **else**
11      Load $O_i^t$ into the SM;
12      **for** $j \leftarrow 1$ **to** $T_{kv}$
13        **if** $\mathcal{J}(\mathcal{S}_s, i, j) == 1$ **then**
14          *# Block-Sparse Skipping*
15          Load $K_j, V_j$;
16          $Q_i K_j^T$;
17          Update $m_{i,j}, \widetilde{P}_{ij}, l_{i,j}$;
18          $\widetilde{P}_{ij} V_j \to O_{i,j}$;
19      $O_i = \mathrm{diag}(l_{i,T_n})^{-1} O_{i,T_n}$
20    Write $O_i$ into HBM;
21 **return** $O = \{O_i\}$;

---

multiplication in TaylorSeer) based on the decoding result. These element-wise operations efficiently update the corresponding features without invoking expensive MMA computations and then assign the results to the output tensor $O$. When a subsequent sparse GEMM-$O$ optimization is applied, the tensor regions no longer require computation, and the corresponding CTAs can even simply return early (Line 7). In the block-sparse skipping branch, an additional reduction-axis decoding function $\mathcal{J}$ (Line 13) is applied before updating $O_{i,j}$ in the inner loop to determine whether to skip specific blocks by decoding $\mathcal{S}_s$. This operation is defined as $\mathcal{J}(\mathcal{S}_s, i, j) = (\mathcal{S}_s >> \frac{i}{n} \frac{T_{kv}}{n} + \frac{j}{n}) \ \& \ 1$. Directly performing this bitwise operation before every $K_j$ tiles loading increases CUDA core overhead. To mitigate this, undecoded bits are processed only once when first encountered, and the results are stored in registers for subsequent reuse, by covering up to $8n$ consecutive blocks.

### 3.5 FLASHOMNI SPARSE GEMM-$Q$/-$O$

Storing redundant computations to avoid recomputation is a standard technique in sequence models (e.g., the KV cache in LLMs, which stores past keys/values so they do not need to be recomputed).

Diffusion, however, is more challenging than token-by-token generation because the reusable tokens do not form a fixed prefix. With FlashOmni-Attention supporting for token-wise feature caching, only a subset of tokens across timesteps is reused, so that the corresponding computations within linear (GEMM) can be skipped. Unlike in LLMs, these cached tokens follow a dynamic, potentially non-contiguous pattern, which is harder to handle. To address this challenge, FlashOmni introduces a sparse GEMM-$Q$/-$O$, to remove redundant computations by utilizing $\mathcal{S}_c$. Moreover, in FlashOmni the kernels are executed in the order GEMM-$Q \rightarrow$ Attention $\rightarrow$ GEMM-$O$, and the cache bias is only incurred in the final GEMM-$O$, while the first two kernels incur no additional memory cost.

Observation 2. In attention modules of common DiT architectures, the derivation of the query vector from $X$ typically involves three sequential operations: query projection ($\text{Proj}_{\text{to\_q}}$), token-wise RMS normalization ($\text{OP}_{\text{RMSNorm}}$), and rotary positional encoding ($\text{OP}_{\text{RoPE}}$). The standard attention output formulation $O_i^h = \sum_j P_{i,j}^h V_j^h$ can be expanded as Equation 2

$$O_i^h = \sum_j \text{Softmax} \left( \frac{\text{OP}_{\text{RoPE}} \left( \text{OP}_{\text{RMSNorm}} \left( X_i W_{to\_q}^h \right) \right) \left( K^h \right)^T}{\sqrt{d}} \right)_j V_j^h \tag{2}$$

Both RMSNorm and RoPE operate exclusively along the feature dimension for each token, without cross-token computation. Therefore, in the sparse step, if $\mathcal{S}_c$ specifies that a particular block $O_i^h$ is retrieved from the cache, the corresponding projection $Q_i^h = X_i W_{to\_q}^h$ can be skipped.

*FlashOmni GEMM-$Q$*. As shown in Figure 3, during the update step, newly computed $Q$ values are required to refresh the symbol information, and the GEMM operation follows its standard execution. In the sparse step, each CTA applies the spatial-axis decoding function $\mathcal{F}$ to $\mathcal{S}_c$ to determine whether its block tile participates in the upcoming attention computation. If not, the CTA exits immediately without performing any further operations.

Observation 3. In typical DiT architectures, once the attention output for each head, $O_i^h$, is obtained, an output projection ($\text{Proj}_{\text{to\_out}}$) is applied, usually to facilitate information exchange across heads for the same token. This operation can be expressed as in Equation 3, where the set $H_i = \{h \mid J(\mathcal{S}_c, i, h) = 1\}$ denotes the heads that actively participate in computation.

$$Out_i = \sum_h O_i^h W_{to\_out}^h = (\sum_{h \notin H_i} + \sum_{h \in H_i}) O_i^h W_{to\_out}^h \tag{3}$$

In the sparse step, when $\mathcal{S}_c$ specifies that the output features of head $h$ for block $i$ are retrieved from cache, the operation can be expressed as $O_i^h = \text{OP}_{\text{reuse}}(\hat{O}_i^h)$ for $h \notin H_i$. Since $\text{OP}_{\text{reuse}}$ is an element-wise operation, the following property holds:

$$\sum_{h \notin H_i} O_i^h W_{to\_out}^h = \sum_{h \notin H_i} \text{OP}_{\text{reuse}}(\hat{O}_i^h) W_{to\_out}^h = \text{OP}_{\text{reuse}}(\sum_{h \notin H_i} \hat{O}_i^h W_{to\_out}^h). \tag{4}$$

This property enables caching of $\sum_{h \notin H_i} \hat{O}_i^h W_{to\_out}^h$ as a bias term $\mathcal{B}_c$ during the update step. In later sparse steps, the cached bias can be transformed using an element-wise kernel, $\text{OP}_{\text{reuse}}(\mathcal{B}_c)$, and added directly to $\text{Proj}_{\text{to\_out}}$. This eliminates redundant reduction-axis computations in the GEMM kernel. Furthermore, storing the cache bias removes the need to retain $\hat{O}_i^h$ in memory during updates. The related element-wise operations in FlashOmni Attention can then be skipped entirely, allowing the cache-then-reuse branch to terminate immediately. This design reduces both computational cost and memory consumption.

*FlashOmni GEMM-$O$*. As illustrated in Figure 3, the GEMM-$O$ computation is divided into two stages. For each block tile $O_i^h$ along the reduction axis, every CTA applies the decoding function $\mathcal{J}$ on $\mathcal{S}_c$ to determine whether the tile should be generated via cache-and-reuse or computed on demand. During the update step, the most recent $\mathcal{S}_c$ and attention outputs $O$ are obtained, and the cache bias $\mathcal{B}_c$ is updated accordingly. In the first stage, CTAs identify tiles that will be reused in the sparse step, compute their outputs, and record the results in $\mathcal{B}_c$. The second stage then relaunches the kernel, pre-initializing the output memory with the cached bias and processing only the tiles that require real-time updates. In the sparse step, the GEMM-$O$ output space is initialized via $\text{OP}_{\text{reuse}}$, executing only the computations corresponding to the second stage of the update step. This strategy eliminates redundant operations on cached tiles, thereby reducing both computational workload and memory usage.

Table 1: End-to-end metrics comparison with block-sparse skipping across image and video generation models.

| Method(seq_len) FLUX.1(4.5K) | Configuration | TOPS (↑) | Sparsity (%, ↑) | PSNR (↑) | LPIPS (↓) | SSIM ↑ | CLIP-IQA (↑) | FID (↓) |
|---|---|---|---|---|---|---|---|---|
| Full-Attention | [dev]: 50 steps | 187.6 | 0 | ∞ | — | — | 0.5154 | — |
|  | 50% steps |  | 50 | 16.69 | 0.3426 | 0.7004 | 0.5054 | 56.589 |
| DiTFastAttnV2 | ($\theta = 0.2$) | 228.5 | 26 | 20.289 | 0.2174 | 0.7901 | 0.5037 | 39.436 |
| SpargeAttn | ($l_1 = 6.5\%, l_2 = 7\%$) | 202.8 | 22 | 21.358 | 0.1868 | 0.8148 | 0.5072 | 34.156 |
| FlashOmni | (50%, 15%, 4, 1, 0%) | 298.5 | 41 | **25.159** | **0.0992** | **0.8838** | 0.5126 | **20.933** |
| FlashOmni | (50%, 15%, 5, 1, 0%) | 304.5 | 43 | 23.994 | 0.1241 | 0.8618 | **0.5165** | 25.594 |
| FlashOmni | (50%, 15%, 5, 2, 30%) | **326.4** | **46** | 23.859 | 0.1281 | 0.8575 | 0.5123 | 25.997 |

| Method(seq_len) Hunyuan Video (33K) | TOPS (↑) | Sparsity (%, ↑) | PSNR (↑) | LPIPS (↓) | SSIM (↑) | Smoothness (↑) | Consistency (↑) | Flickering (↑) | Style (↑) |
|---|---|---|---|---|---|---|---|---|---|
| Full-Attention (50 steps) | 184.6 | 0 | ∞ | — | — | 99.33 | 97.87 | 99.221 | 0.2394 |
| DiTFastAttnV2 ($\theta = 0.2$) | 249.7 | 31 | 19.751 | 0.3086 | 0.7331 | 99.13 | 97.44 | 99.098 | 0.2381 |
| SpargeAttn (untuned: topk=73%) | 205.7 | 27 | 23.57 | 0.1919 | 0.8176 | 99.3 | 97.58 | 99.152 | 0.2421 |
| SpargeAttn ($l_1 = 6\%, l_2 = 6.5\%$) | 221.1 | 32 | 21.701 | 0.2442 | 0.7748 | 99.21 | 97.42 | 98.993 | 0.2399 |
| FlashOmni (40%, 1%, 3, 1, 0) | 271.9 | 34 | **32.192** | **0.0571** | **0.9289** | 99.32 | 97.69 | **99.217** | 0.2401 |
| FlashOmni (40%, 1%, 6, 1, 0) | 299.7 | 39 | 28.733 | 0.0946 | 0.8918 | 99.29 | 97.55 | 99.198 | 0.2401 |
| FlashOmni (50%, 5%, 6, 1, 30%) | **346.2** | **47** | 27.877 | 0.1195 | 0.8699 | **99.33** | **97.88** | 99.224 | 0.2392 |

# 4 EXPERIMENTS

## 4.1 SETUP

**Models**. Our experiments are performed on three SOTA visual generative models: the text-to-image generator FLUX.1-dev (Black-Forest-Labs, 2024), the text-to-video generator HunyuanVideo (Kong et al., 2024), and the text-guided image editing model FLUX.1-Kontext (Batifol et al., 2025).

**Benchmarks and Metrics**. For the text-to-image generation experiments, we conduct inference on the COCO-2017 dataset (Lin et al., 2014) at a resolution of $1024 \times 1024$ pixels. The text-guided image editing evaluation is performed on KontextBench (Batifol et al., 2025), which contains 1,026 unique image–prompt pairs derived from 108 base images. Generated outputs are assessed using CLIP-IQA (Wang et al., 2023) and FID-FP16 (Heusel et al., 2017), measuring image quality and distributional similarity, respectively. For text-to-video generation, we adopt VBench (Huang et al., 2024) to evaluate various aspects of video quality, including background consistency, motion smoothness, temporal flicker, and stylistic coherence. Fidelity to the original results is quantified using PSNR, SSIM (Wang et al., 2004), and LPIPS (Zhang et al., 2018).

To measure computational efficiency of sparse attention, we report TOPS (tera-operations per second), defined as `attn/t`, and **Sparsity**, defined as (`skip`/`total`) used in SpargeAttn (Zhang et al., 2025b). Here, `attn` denotes the total number of operations in a standard attention computation, `t` is the latency from given $(Q, K, V)$ inputs to the corresponding attention outputs, `skip` is the number of skipped attention pairs $(Q_i K_j^\top, \widetilde{P}_{ij} V_j)$, and `total` is the total number of such pairs.

**Baselines and Platform**. We compare FlashOmni with five SOTA baselines: two block-sparse attention skipping methods: SpargeAttn (Zhang et al., 2025b) and DiTFastAttnV2 (Zhang et al., 2025a), and three feature caching approaches: ToCa (Zou et al., 2025), FORA (Selvaraju et al., 2024), and TaylorSeer (Liu et al., 2025b). The configuration for FlashOmni is specified as $(\tau_q, \tau_{kv}, \mathcal{N}, \mathcal{D}, S_q)$ (Appendix A.3.2). All experiments are executed on NVIDIA A100 and RTX 4090.

## 4.2 VISUAL QUALITY EVALUATION

As shown in Table 1 and Table 2, FlashOmni consistently outperforms all baselines across quality metrics on FLUX and HunyuanVideo. Compared with block-sparse skipping with tuning methods (DiTFastAttnV2, SpargeAttn), it achieves greater sparsity while clearly maintaining superior quality. Comparing against the SpargeAttn with untuned setting, and our method still delivers better performance with higher sparsity. Against caching methods (FORA, ToCa, TaylorSeer), FlashOmni attains better quality on FLUX under the same moderate cache interval ($\mathcal{N} = 5$) and order $\mathcal{D}$. Even under the more aggressive interval ($\mathcal{N} = 6$), FlashOmni maintains substantial advantages on both FLUX (1.74 ↑ SSIM over TaylorSeer) and HunyuanVideo (2.97 ↑ SSIM over TaylorSeer). Similar performance gains are also observed on FLUX.1-Kontext (Table 6), where FlashOmni surpasses all other methods.

Table 2: End-to-end metrics comparison with feature caching across image and video generation models.

| Method(seq_len) FLUX.1(4.5K) | Configuration | PSNR (↑) | LPIPS (↓) | SSIM ↑ | CLIP-IQA (↑) | FID (↓) |
|---|---|---|---|---|---|---|
| Full-Attention | [dev]: 50 steps | ∞ | — | — | 0.5154 | — |
| | 50% steps | 16.69 | 0.3426 | 0.7004 | 0.5054 | 56.589 |
| FORA | $(\mathcal{N}=5)$ | 22.846 | 0.1595 | 0.825 | 0.5111 | 31.188 |
| ToCa | | 22.827 | 0.2059 | 0.7978 | 0.5095 | 29.947 |
| TaylorSeer | $(\mathcal{N}=5, \mathcal{D}=1)$ | 22.852 | 0.1441 | 0.8361 | 0.5103 | 28.253 |
| TaylorSeer | $(\mathcal{N}=5, \mathcal{D}=2)$ | 23.177 | 0.1402 | 0.8395 | 0.5131 | 27.376 |
| FlashOmni | $(50\%, 15\%, 5, 1, 30\%)$ | 23.866 | 0.1283 | 0.8575 | 0.5119 | 25.994 |
| FlashOmni | $(50\%, 15\%, 5, 2, 30\%)$ | 23.859 | 0.1281 | 0.8575 | 0.5123 | 25.997 |
| FlashOmni | $(50\%, 15\%, 5, 1, 0\%)$ | **23.994** | **0.1241** | **0.8618** | **0.5165** | **25.594** |
| TaylorSeer | $(\mathcal{N}=6, \mathcal{D}=2)$ | 22.245 | 0.1615 | 0.8199 | 0.5119 | 31.082 |
| FlashOmni | $(50\%, 15\%, 6, 1, 30\%)$ | **23.217** | **0.1507** | **0.8373** | **0.5124** | **28.965** |

| Method(seq_len) Hunyuan Video (33K) | PSNR (↑) | LPIPS (↓) | SSIM (↑) | Smoothness (↑) | Consistency (↑) | Flickering (↑) | Style (↑) |
|---|---|---|---|---|---|---|---|
| Full-Attention (50 steps) | ∞ | — | — | 99.33 | 97.87 | 99.221 | 0.2394 |
| FORA $(\mathcal{N}=6)$ | 26.702 | 0.1351 | 0.8539 | 99.31 | 97.89 | 99.221 | 0.2382 |
| TaylorSeer $(\mathcal{N}=6, \mathcal{D}=1)$ | 26.483 | 0.1229 | 0.8621 | 99.32 | 97.38 | 99.196 | 0.2388 |
| FlashOmni $(50\%, 5\%, 6, 1, 30\%)$ | 27.877 | 0.1195 | 0.8699 | **99.33** | 97.88 | **99.224** | 0.2392 |
| FlashOmni $(40\%, 1\%, 6, 1, 0)$ | **28.733** | **0.0946** | **0.8918** | 99.29 | 97.55 | 99.198 | **0.2401** |

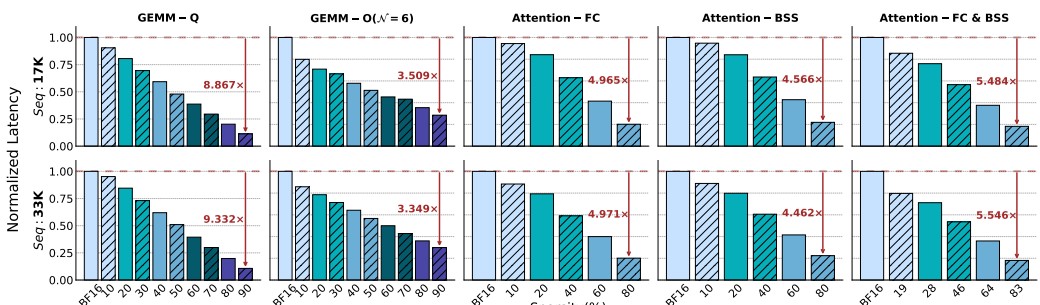

Figure 5: Normalized kernel latency on NVIDIA A100 (BF16, compared with cuBlas and FlashInfer) at different sparsity levels for sparse GEMM-$Q$/-$O$ ($\mathcal{N}=6$) and Attention kernels. FC and BSS indicate the application of feature caching and block-sparse skipping, respectively.

## 4.3 EFFICIENCY EVALUATION

**Attention performance.** We implement our sparse attention kernel based on FlashInfer (Ye et al., 2025) for half-precision and SageAttention (Zhang et al.) for 8-bit computation, and optimize the storage and decoding overhead within attention kernel introduced by these sparse symbols. We evaluate its performance through a systematic comparison with relevant baselines using randomly generated sparse symbols under three configurations: (1) feature caching (FC) only, (2) block-sparse skipping (BSS) only, and (3) both enabled. For each configuration, we report speedup across varying sparsity ratios and compare it with the theoretical computation reduction. The right three columns in Figure 5 show that the empirical speedup closely matches the theoretical prediction and scales linearly with sparsity.

At the same sparsity level, FC consistently yields higher performance than BSS. For instance, at 80% sparsity, FC achieves a 4.97× speedup, while BSS reaches 4.6×. This difference arises because FC requires decoding only once per CTA, whereas BSS performs decoding repeatedly throughout the reduction process, incurring additional CUDA core overhead. More detailed experimental results about attention kernel performance are provided in the Appendix A.4.

**Sparse GEMMs performance.** We also implement our sparse GEMM kernels based on CuTe to mitigate redundant computation and memory overhead introduced by enabling the FC strategy in attention. We conduct extensive experiments on both the GEMM-$Q$/-$O$, observing a consistent linear speedup trend with increasing sparsity. For GEMM-$Q$, acceleration occurs along the spatial axis. Since decoding is performed only once, the achieved speedup closely matches the theoretical upper bound of computation reduction. For GEMM-$O$, sparsity lies along the reduction axis, requiring

Table 3: End-to-end speedup comparison for HunyuanVideo generation on A100 with a sequence length of 33K. Latency is measured in seconds, using the same generation settings as in the visual quality evaluation.

| Method: HunyuanVideo(33K) | Sparsity (%, ↑) | End-to-end (s, ↓) | Speedup (↑) | Attention (s, ↓) | Preprocess (↓) |
|---|---|---|---|---|---|
| Diffusers-FlashAttentionV2 | 0 | 419 | 1 | 218 | 0 |
| SpargeAttn ($l_1 = 6\%$, $l_2 = 6.5\%$) | 32 | 401 | 1.05 | 171.5 | $\sim 30$ |
| DiTFastAttnV2 ($\theta = 0.2$) | 31 | 332 | 1.26 | 131 | 0 |
| FlashOmni (50%, 5%, 6, 1, 30%) | 47 | 278 | 1.51 | 109.1 | $\sim 0.4$ |

multiple decoding operations, preventing a one-to-one match with the theoretical peak speedup; nevertheless, the achieved performance remains close. For instance, at 90% sparsity with $\mathcal{N} = 6$, the theoretical speedup is $4\times$, while our kernel attains $3.3\times \sim 3.5\times$ (Figure 5). This discrepancy arises because GEMM-$O$ requires multiple decoding operations along the reduction axis on CUDA cores; even at 90% sparsity, while GEMM-$Q$ alone can achieve a near-linear $9\times$ speedup, GEMM-$O$ only reaches $6.39\times$ due to these additional overheads. More details refer to the Appendix A.3.3 and A.5.

**Density analysis.** We further measure the computation density during inference on HunyuanVideo, as illustrated in Figure 6. For SpargeAttn, the density remains relatively stable throughout the inference process. In contrast, FlashOmni exhibits a sharp drop in density from an initial value close to 1, maintaining overall lower density both at the whole-model level and within individual transformer layers.

We attribute FlashOmni's ability to achieve higher video generation quality despite lower density to the critical role of the early denoising steps in the DiT inference process. In these initial stages, the vision token is randomly initialized from Gaussian noise and requires strong and comprehensive text guidance, alongside timely updates of the textual signal, consistent with our findings in Observation 1. Moreover, under our degradation strategy, when the proportion of token blocks requiring updates falls below a certain threshold, we directly employ full-feature caching to improve efficiency, whose details refer to Appendix A.3.2.

**Memory Overhead.** We analyze the memory cost for sparse symbols and cache bias used in FlashOmni. Let $H$ be the number of heads, $\text{Size}_f$ the feature-caching symbol size, $\text{Size}_b$ the block-skipping symbol size, $N$ the number of sequence length, and $L$ the number of transformer

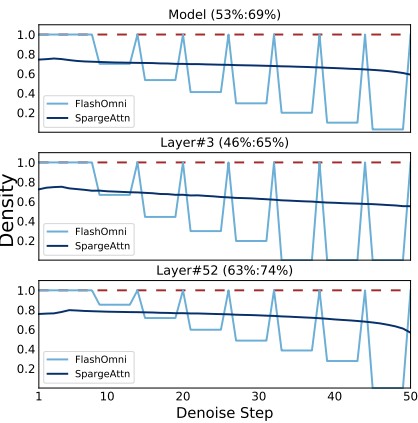

Figure 6: Density comparison with SpargeAttn on HunyuanVideo benchmark.

blocks. Our method stores $\mathcal{S}_s = LH(\frac{N}{\text{Size}_f} \times \frac{N}{\text{Size}_b})/8$ bytes and $\mathcal{S}_c = LH(\frac{N}{\text{Size}_f})/8$ bytes for sparse symbols. For example, with $\text{Size}_f = \text{Size}_b = 128$ on HunyunVideo and video resolution $544 \times 960 \times 61$ (33K tokens) in our generation setting, our sparse symbols occupy about 11MB, whereas SpargeAttn's LUT consumes over 700MB, resulting in more than 60× memory reduction; for even higher-resolution video generation, this symbol overhead remains tiny compared with the overall model and activation memory and can be regarded as negligible. For cache bias, if feature caching is disabled in FlashOmni, there is no extra memory cost, matching pure block-skipping methods. FlashOmni only runs the attention kernel, without invoking GEMM-$Q$/-$O$, and thus without any caching. When feature caching is enabled, FlashOmni executes kernels in the order GEMM-$Q \rightarrow$ Attention $\rightarrow$ GEMM-$O$. Only the final GEMM-$O$ uses caching with the same size as in other feature-caching approaches, while GEMM-$Q$ and Attention incur no memory overhead.

**End-to-End Inference Comparison.** We report the end-to-end performance comparison on the HunyuanVideo against recent SOTA sparse attention kernels on A100, with using the same generation setting in Table 1, as shown in the Table 3. SpargeAttn uses a dynamic attention kernel but suffers from insufficient computation and heavy preprocessing. At the same time, DitFastAttnV2 is a static sparse kernel that avoids extra overhead once fixed, achieving higher efficiency at the same sparsity but with lower flexibility and generation quality. Our kernel also supports dynamic attention, with lower memory overhead and preprocessing cost than SpargeAttn, leading to stronger performance. It further overcomes prior limitations of only supporting block skipping or feature caching within a single layer, enabling more flexible combinations of sparse strategies for better adaptability and generation quality. FlashOmni delivers up to $1.5\times$ end-to-end acceleration under a 47% sparsity, while maintaining higher generation quality, shown in Table 1.

Table 4: The ablation study evaluates FlashOmni with different configurations on FLUX.1.

| Configuration | Sparsity (%, ↑) | PSNR (↑) | LPIPS (↓) | SSIM (↑) | FID (↓) |
|---|---|---|---|---|---|
| $(5\%, 15\%, \mathcal{N} = 3, 1, 0)$ | 24% | 26.702 | 0.0784 | 0.9041 | 16.495 |
| $(5\%, 15\%, \mathcal{N} = 4, 1, 0)$ | 26% | 25.217 | 0.0999 | 0.8837 | 20.789 |
| $(5\%, 15\%, \mathcal{N} = 5, 1, 0)$ | 28% | 24.193 | 0.1214 | 0.8650 | 24.860 |
| $(5\%, 15\%, \mathcal{N} = 6, 1, 0)$ | 29% | 23.454 | 0.1366 | 0.8506 | 28.072 |
| $(5\%, 15\%, \mathcal{N} = 7, 1, 0)$ | 30% | 23.012 | 0.1488 | 0.8405 | 30.541 |
| $(50\%, 15\%, 5, \mathcal{D} = 0, 30\%)$ | 46% | 23.355 | 0.1452 | 0.8452 | 28.925 |
| $(50\%, 15\%, 5, \mathcal{D} = 1, 30\%)$ | 46% | 23.866 | 0.1283 | 0.8575 | 25.994 |
| $(50\%, 15\%, 5, \mathcal{D} = 2, 30\%)$ | 46% | 23.859 | 0.1281 | 0.8575 | 25.997 |
| $(\tau_q = 5\%, 15\%, 4, 1, 0)$ | 28% | 25.217 | 0.0999 | 0.8838 | 20.789 |
| $(\tau_q = 20\%, 15\%, 4, 1, 0)$ | 36% | 25.207 | 0.0989 | 0.8841 | 20.881 |
| $(\tau_q = 35\%, 15\%, 4, 1, 0)$ | 40% | 25.179 | 0.0991 | 0.8838 | 20.913 |
| $(\tau_q = 50\%, 15\%, 4, 1, 0)$ | 41% | 25.159 | 0.0992 | 0.8838 | 20.933 |

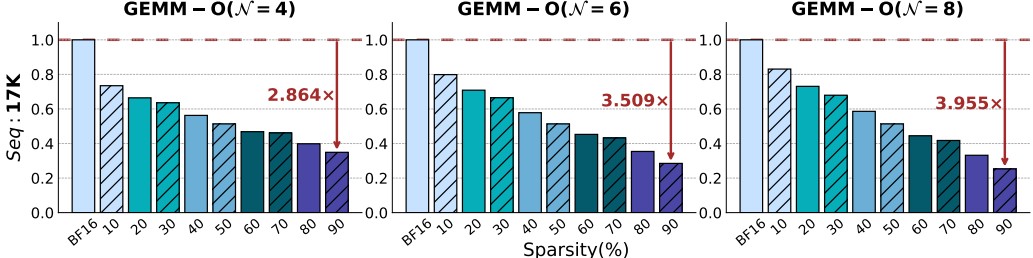

Figure 7: Normalized kernel latency at different $\mathcal{N}$ for sparse GEMM-$O$, where token length is 17K.

## 4.4 ABLATION STUDY

We perform ablation experiments (Table 4) on Flux to assess how the interval $\mathcal{N}$ and caching order $\mathcal{D}$ affect generation quality. First-order caching ($\mathcal{D} = 1$) delivers the largest performance gains, while higher orders degrade quality, highlighting the limits of simulation in capturing real-world behavior. Despite its bias, first-order caching remains more faithful than direct reuse. Increasing $\mathcal{N}$ offers greater acceleration but at the cost of noticeable quality loss. We also conduct an ablation on sparsity, showing that generation quality remains largely stable across configurations. This robustness comes from: (1) a sparsity schedule that keeps sparsity low in early steps and gradually increases it later; and (2) the strategy in (3.3), which focuses on the most important image–text token pairs, with the CDF-based mechanism consistently updating the most critical blocks at the FC level. We also found that when $N$ becomes large, generation quality still degrades, even if the sparsity is lower than that of smaller $N$ settings with higher sparsity. This is because a larger $N$ results in longer intervals during which the selected cached tokens are not updated with precise attention, causing errors to accumulate over time.

We further conduct ablation experiments (Figure 7) to evaluate the speedup of GEMM-$O$ across settings $\mathcal{N}$ with a sequence length of 17K. As $\mathcal{N}$ increases, the overall speedup consistently improves but falls short of the theoretical speedup predicted by reduced FLOPs. For example, when $\mathcal{N} = 4, 6, 8$, the measured 93.1%, 87.7%, and 84.7% of theoretical speedup, respectively. This discrepancy arises because GEMM-$O$ requires multiple decoding operations along the reduction axis on CUDA cores; even at 90% sparsity, while GEMM-$Q$ alone can achieve a near-linear $9\times$ speedup, GEMM-$O$ only reaches $6.39\times$ due to these additional overheads. More details refer to the Appendix A.5.

## 5 CONCLUSION

We propose FlashOmni, a unified sparse attention engine that is applicable to any Diffusion Transformers. FlashOmni abstracts an *Update–Sparse* paradigm with three key designs: it unifies multi-granularity spare strategies with flexible sparse symbols, supports efficient arbitrary sparse attention through a general kernel, and optimizes sparse GEMMs in attention linear layers to remove redundant computation. Experiments demonstrate that FlashOmni achieves significant end-to-end acceleration across diverse models without retraining, while preserving generation quality.

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

# A APPENDIX

## A.1 THE USE OF LARGE LANGUAGE MODELS (LLMs)

We employed a large language model (LLM) to assist in language refinement during manuscript preparation.

## A.2 ETHICS STATEMENT

The authors have read and complied with the ICLR Code of Ethics. This research does not involve human participants, personal or sensitive information, and does not pose foreseeable risks of harm.

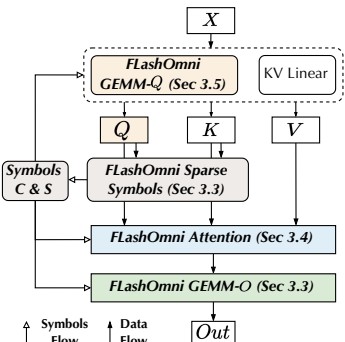

Figure 8: FlashOmni design.

| Settings | Description |
|---|---|
| $\tau_q$ | Sparsity threshold for $q$. Details: the importance scores of tokens are sorted in ascending order, and tokens are progressively marked for sparsification until the cumulative importance of the selected tokens exceeds $\tau_q$. |
| $\tau_{kv}$ | Sparsity threshold for kv. Details: the importance scores of blocks are sorted in ascending order, and blocks are progressively marked for sparsification until the cumulative importance of the selected blocks exceeds $\tau_{kv}$. |
| $\mathcal{N}$ | Moderate cache interval. |
| $\mathcal{D}$ | Order of expansion. |
| $S_q$ | Threshold of caching. Details: if the proportion of tokens requiring computation is below this threshold, the layer degenerates into feature caching. |

Table 5: Settings and their descriptions.

## A.3 FLASHOMNI DETAILS

### A.3.1 FLASHOMNI DESIGN

Figure 8 shows the system overview of FlashOmni, a unified sparse attention engine for Diffusion Transformers. FlashOmni's design consists of three key components: FlashOmni Sparse symbols 3.3, Attention 3.4, GEMM-$Q$/-$O$ 3.5, each targeting one of the major computation kernels in a Transformer block and applying tailored sparsity optimizations to them.

### A.3.2 EXPERIMENTAL SETUP

As is metioned in 4.1, the configuration for FlashOmni is specified as $(\tau_q, \tau_{kv}, \mathcal{N}, \mathcal{D}, S_q)$. In Table 5, we give details about them. It is worth noting that the values of $\tau_q$ and $\tau_{kv}$ are not set to their target values at the initial step. Rather, they progressively converge to these values as the time step advances.

We conducted experiments on three models for different tasks, FLUX for the text-to-image generator, HunyuanVideo for text-to-video generator, FLUX.1-Kontext for text-guided image editing. We selected five representative parameters for evaluation, including $\tau_q$ at 5% and 50%; $\tau_{kv}$ at 15%, $\mathcal{N}$ of 3, 4, 5, 6, and 7; $\mathcal{D}$ of 0, 1, and 2; $S_q$ at 0% and 30%. Our experiments achieved excellent results, maintaining high generation quality even under high sparsity levels. It is worth noting that these parameters can be efficiently tuned via lightweight search algorithms to further enhance the performance of FlashOmni. We plan to implement this optimization in future work.

### A.3.3 GEMM-$O$ SPEEDUP METRICS

In this section, we define our methodology for computing the speedup metric of GEMM-$O$. Define $T_{total}$ as the total time taken to execute a single $\mathrm{Proj}_{\mathrm{to\_out}}$ operation and $s$ to denote the sparsity ratio, then we can compute the normal time consumption: $\mathcal{N}T_{total}$, and the FlashOmni time consumption:

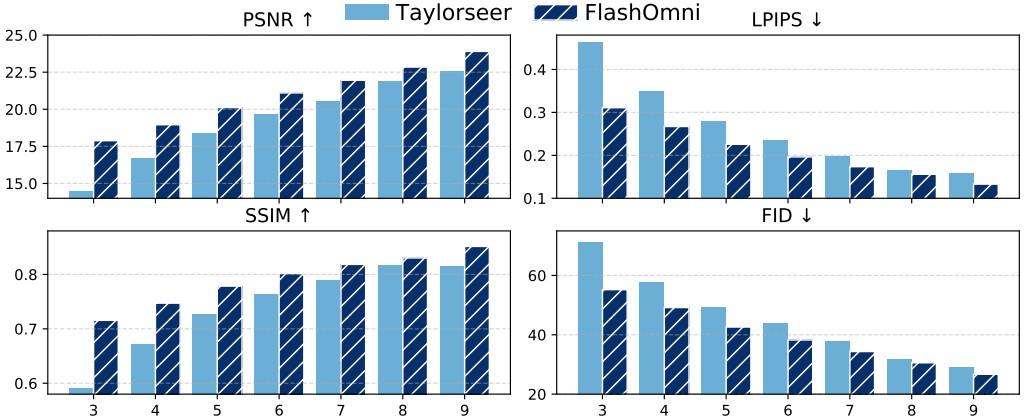

Figure 9: End-to-end metrics comparison with feature caching on FLUX with different warmup steps.

Table 6: End-to-end metrics comparison on text-guided image editing model FLUX-Kontext.

| Method FLUX.1-Kontext | Configuration | PSNR (↑) | LPIPS (↓) | SSIM ↑ | FID (↓) |
|---|---|---|---|---|---|
| DiTFastAttnV2 | $(\theta = 0.2)$ | 24.507 | 0.1233 | 0.8225 | 37.232 |
| SpargeAttn | $(l_1 = 6\%, l_2 = 6.5\%)$ | 26.851 | 0.1048 | 0.8519 | 28.163 |
| FlashOmni | $(50\%, 15\%, 5, 1, 0)$ | **31.590** | **0.0466** | **0.9201** | **13.218** |
| TaylorSeer | $(\mathcal{N} = 5, \mathcal{D} = 1)$ | 29.733 | 0.062 | 0.8919 | 15.876 |
| FlashOmni | $(50\%, 15\%, 5, 1, 20\%)$ | **30.689** | **0.0543** | **0.9082** | **15.204** |

$$T_{\text{FlashOmini}} = T_{\text{sparse}} + T_{\text{computation}}$$
$$= T_{\text{total}} \times s + \sum_{\mathcal{N}} \left( (1 - s) \times T_{\text{total}} \right)$$
$$= T_{\text{total}} + (\mathcal{N} - 1)(1 - s)T_{\text{total}} \tag{5}$$

Then we can compute the speedup metric $\frac{\mathcal{N}}{1+(\mathcal{N}-1)(1-s)}$. When the $s = 0.9$ and $\mathcal{N} = 6$, the theoretical speedup is $\frac{6}{1+(6-1)(1-0.9)} = 4$. As is shown in Figure 5, our actual speedup obtained under this configuration is $3.509\times$, approaching the theoretical value of $4\times$, which substantiates the high efficiency of FlashOmni.

### A.3.4 WARMUP STEPS ANALYSIS FOR FLUX

Figure 9 shows the end-to-end metrics comparison with feature caching on FLUX with different warmup steps. It is evident that when the warmup step count is low, the TaylorSeer method experiences a significant drop in image quality, with poor performance across PSNR, LPIPS, SSIM, and FID metrics, indicating a strong dependence on a high number of warmup steps. In contrast, while the FlashOmni method also shows a decline in quality at lower warmup steps, it still maintains relatively high generation quality and does not require a large warmup step count to achieve satisfactory results.

### A.3.5 SUPPLEMENTARY FLUX.1-KONTEXT RESULTS

Table 6 complements the evaluation by providing an end-to-end metrics comparison on a text-guided image editing model. The results demonstrate the superior performance of FlashOmni in the FLUX.1-Kontext setting for text-guided image editing.

### A.4 FLASHOMNI ATTENTION

We evaluate the inference performance of the attention kernel under different sparsity levels during high-resolution diffusion, including 2K image generation in Flux, video generation in HunyuanVideo, using sequence lengths of 17K and 33K with 24 heads and 128 headdim as examples. Three configurations are tested: activating only FC, activating only BSS, and activating both. All sparse symbols are randomly generated, with the random seeds for BSS varying across groups. @1, @2, and

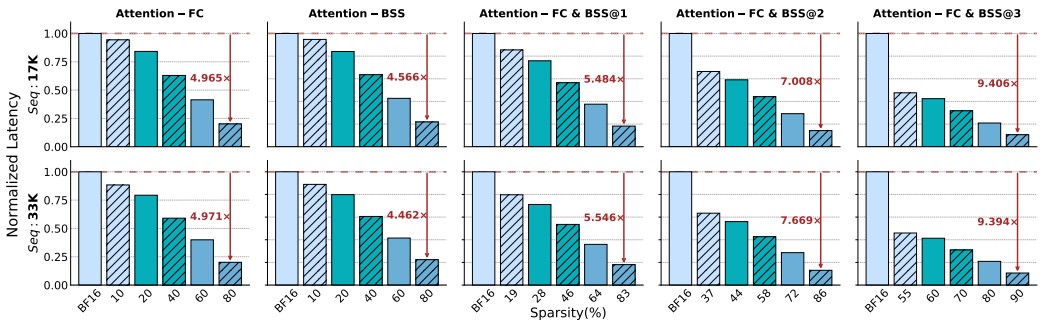

Figure 10: Normalized kernel latency on NVIDIA A100 (BF16, compared with FlashInfer) at different sparsity levels for sparse attention kernels. FC and BSS indicate the application of feature caching and block-sparse skipping, respectively.

@3 correspond to BSS sparsity thresholds of 0.1, 0.3, and 0.5, respectively, while, within each group, the FC threshold increases incrementally (0.1, 0.2, 0.4, 0.6, 0.8). Our experiments show that, at the same sparsity level, FC achieves higher speedup than BSS. This is because FC requires only a single decoding operation on the CUDA cores, whereas BSS—despite having the same sparsity—requires multiple decoding operations along the reduction axis, which reduces efficiency. For the combined sparsity strategy, the speedup scales almost linearly with sparsity (approximately a 1:1 ratio). As an illustration, a combined sparsity of 90% yields an observed speedup of about $\sim 9.4\times$ in Figure 10.

Table 7: Scaling analysis of FlashOmni Attention across varying sequence lengths on NVIDIA A100.

| Sequence Length | FlashAttentionV2 (ms) | Sparsity | FlashOmni Attention (ms) | Speedup | Note |
|---|---|---|---|---|---|
| 2,560 | 0.434 | 23% | 0.356 | 1.22 | 1024× 512 image |
| 4,608 | 1.264 | 64% | 0.604 | 2.09 | 1024× 1024 image |
| 9,750 | 6.62 | 43% | 3.87 | 1.71 | random config |
| 16,896 | 17.99 | 58% | 8.28 | 2.17 | 2049× 2048 image |
| 20,480 | 26.6 | 23% | 22.05 | 1.21 | random config |
| 25,152 | 42.6 | 43% | 25.06 | 1.7 | random config |
| 33,152 | 69.62 | 64% | 27.54 | 2.53 | 544×960 ×61 video |
| 43,890 | 128.7 | 43% | 74.79 | 1.72 | random config |
| 54,130 | 196.2 | 43% | 114.8 | 1.71 | random config |
| 66,048 | 276 | 46% | 162 | 1.71 | 1024×1024 ×61 video |

We report an additional scaling analysis within the ablation experiments to evaluate the efficiency of the FlashOmni Attention kernel with 24 heads and 128 head dimensions across varying sequence lengths, as shown in Table 7. FlashOmni consistently delivers a stable speedup over the baselines, indicating that the performance gains are robust and do not degrade as the problem size scales.

We also benchmark kernel performance on an RTX 4090 at matched sparsity levels (since the SpargeAttn top-k API cannot represent the feature-caching sparsity used in FlashOmni) for different head dimensions (e.g., 64 and 128). The results are measured in milliseconds and reported in Table 8. To ensure a fair comparison with SpargeAttn, we implemented an 8-bit kernel based on SageAttention (Zhang et al.). Under identical sparsity settings, our method achieves comparable computational efficiency, demonstrating that our design does not compromise compute performance. More importantly, our approach aggressively optimizes the memory footprint of sparse symbols, and the sparse-symbol design enables a single kernel to simultaneously support both feature-caching and block-skipping sparsity.

Table 8: Attention performance comparison on RTX 4090 with 24 heads and different head dimensions.

| Seq Length = 4608, head_dim = 128 | | | Seq Length = 16896, head_dim = 128 | | | Seq Length = 16896, head_dim = 64 | | |
|---|---|---|---|---|---|---|---|---|
| Sparsity | SpargeAttn | FlashOmni | Sparsity | SpargeAttn | FlashOmni | Sparsity | SpargeAttn | FlashOmni |
| 14% | 0.613 | 0.551 | 14% | 7.621 | 7.302 | 14% | 4.48 | 4.554 |
| 20% | 0.573 | 0.550 | 28% | 6.396 | 5.985 | 28% | 3.835 | 3.772 |
| 37% | 0.461 | 0.483 | 37% | 5.594 | 5.501 | 37% | 3.354 | 3.387 |
| 44% | 0.419 | 0.409 | 44% | 4.960 | 4.725 | 44% | 2.974 | 2.986 |
| 55% | 0.339 | 0.342 | 52% | 4.26 | 4.235 | 52% | 2.553 | 2.559 |
| 60% | 0.300 | 0.305 | 64% | 3.206 | 3.209 | 64% | 1.932 | 1.963 |

Table 9: Performance analysis of FlashOmni GEMM-*Q*/-*O* with different sparsity on NVIDIA A100 .

| Seq length: 17K | cuBlas | GEMM-$Q$ | | cuBlas | GEMM-$O$ | |
|---|---|---|---|---|---|---|
| Sparsity | Latency | Latency | Speedup ($\uparrow$) | Latency | Latency | Speedup ($\uparrow$) |
| 0% | | 1.371 | 97.5% | | 1.512 | 91.1% |
| 40% | 1.337 | 0.797 | 167.8% | 1.377 | 0.941 | 146.3% |
| 60% | | 0.525 | 254.7% | | 0.661 | 208.3% |

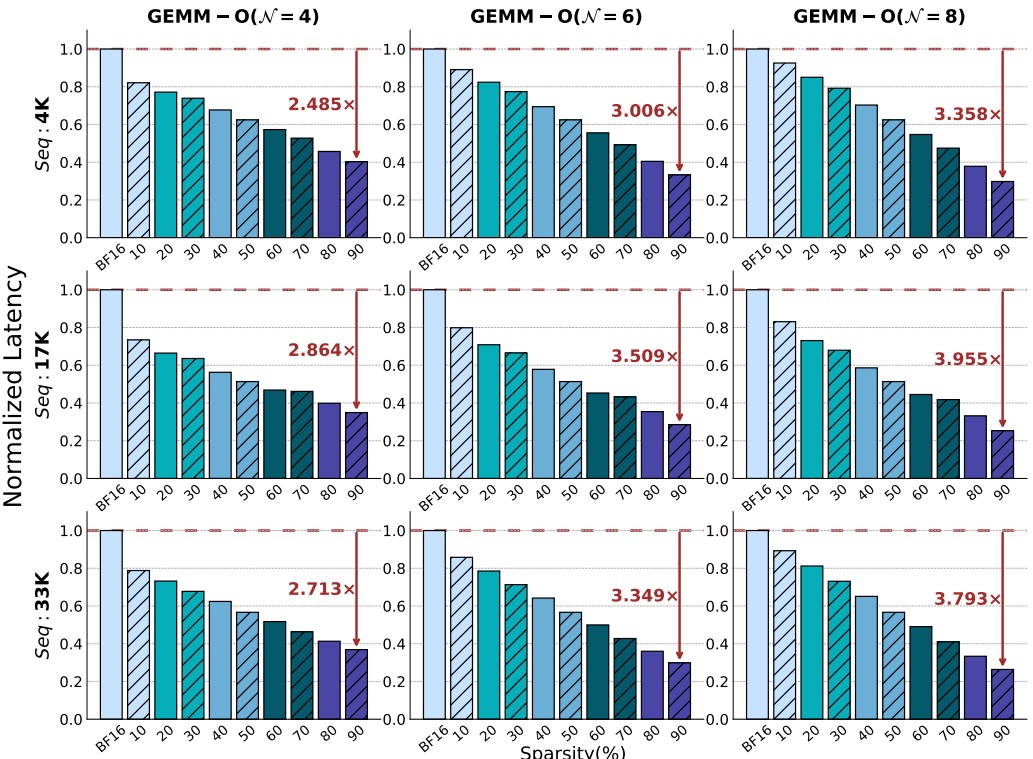

Figure 11: Normalized kernel latency on NVIDIA A100 (BF16) at different sparsity levels for sparse GEMM-*O*, whose $\mathcal{N}$ includes 4,6,8.

## A.5 FLASHOMNI SPARSE GEMM-*Q*/-*O*

We conduct a performance analysis of sparse GEMM-*Q*/-*O* using a sequence length of 16,896 and a weight matrix of size 4096×3072, which is a typical configuration in modern Transformer blocks. The results are measured in milliseconds and reported in the Table 9, even without sparsity, the performance gap between GEMM-*Q* and GEMM-*O* is already visible. Compared with GEMM-*Q*, where the sparsity judgement on CUDA cores is performed only once, GEMM-*Q* incurs negligible extra overhead and thus can achieve nearly linear theoretical speedup. In contrast, the main bottleneck in GEMM-*O* arises from decoding operations. Specifically, the sparsity judgement for GEMM-O occurs along the reduction axis, resulting in bitwise operations multiple times on CUDA cores, thereby reducing the proportion of MMA computations in the overall kernel against GEMM-*Q*.

We further evaluated the GEMM-*O* speedup performance across three representative resolution settings for the generation tasks. As shown in the Figure 11, for tasks at standard resolutions, the parallelism of the kernel is relatively limited, and the influence of decoding operations becomes higher. Consequently, the speedup is lower compared to the ultra-high-resolution scenarios. Nevertheless, the acceleration remains notable, achieving approximately 2.5×−3.6× speedup across different $\mathcal{N}$ settings. In the ultra-high-resolution cases, the speedup increases further, reaching 2.7×−3.9×.

## A.6   FLASHOMNI PROGRAMMING INTERFACE

```python
import flashomni

# Attention Processor Modules
def __attn_wrapper__(self = AttnProcessor, task_info):
    self.attn_proc = flashomni.AttentionWrapper(task_info)

def __call__(self = AttnProcessor, attn, x, cache_dic):
  q = flashomni.to_q(cache_dic.sparse_symbols, x)                    # FlashOmni GEMM Q
  ...
  attn_out = self.attn_proc(q,k,v, cache_dic.sparse_symbols)         # FlashOmni Attention
  ...
  if cache_dic[type] == "update":
    # FlashOmni Sparse Symbols
    cache_dic.sparse_symbols = self.update_sparse_symbols(q, k)
    # Cache bias generation
    cached_bias = flashomni.to_out(attn_out, cache_dic.sparse_symbols)   # FlashOmni GEMM O

  out = flashomni.to_out(attn_out, cache_dic.sparse_symbols, cached_bias)  # FlashOmni GEMM O
  return out
```

## A.7 SUPPLEMENTARY VISUALIZATION RESULTS

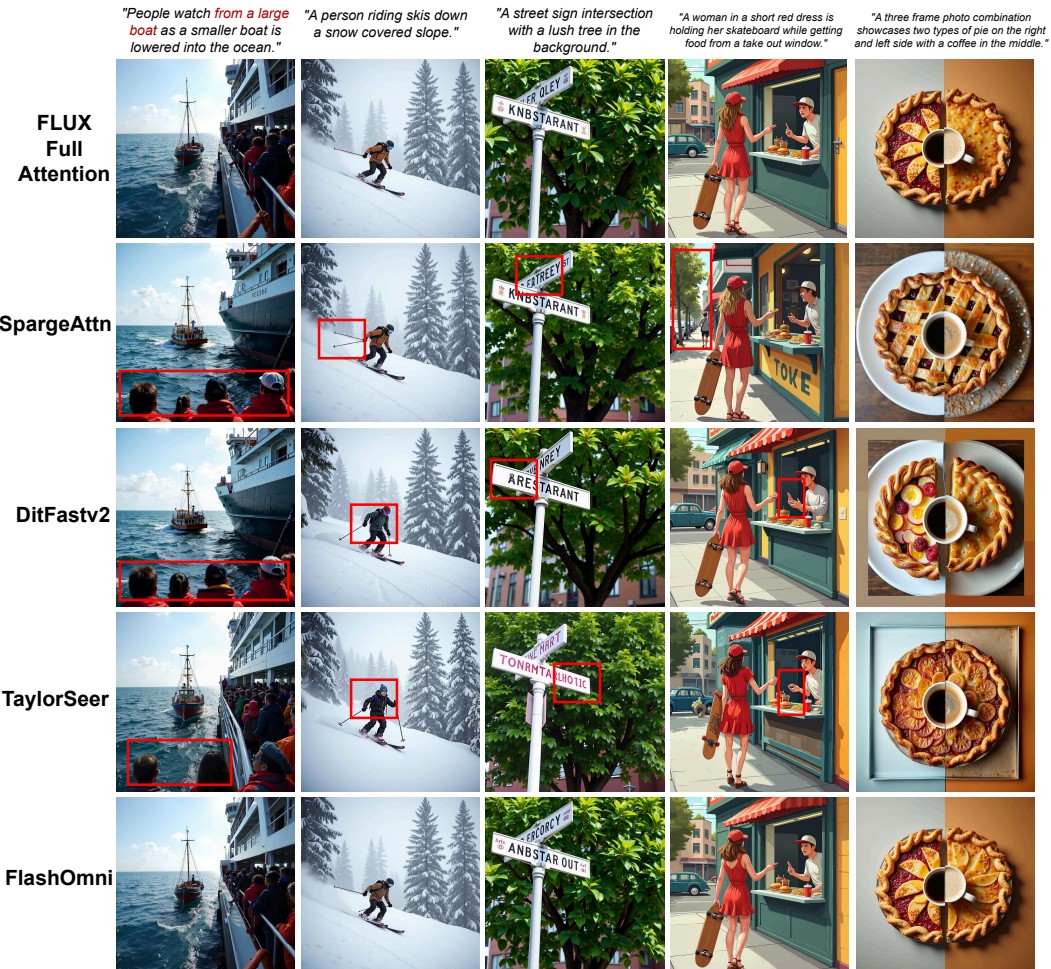

Figure 12: Visualization results for different acceleration methods on FLUX.1-dev.

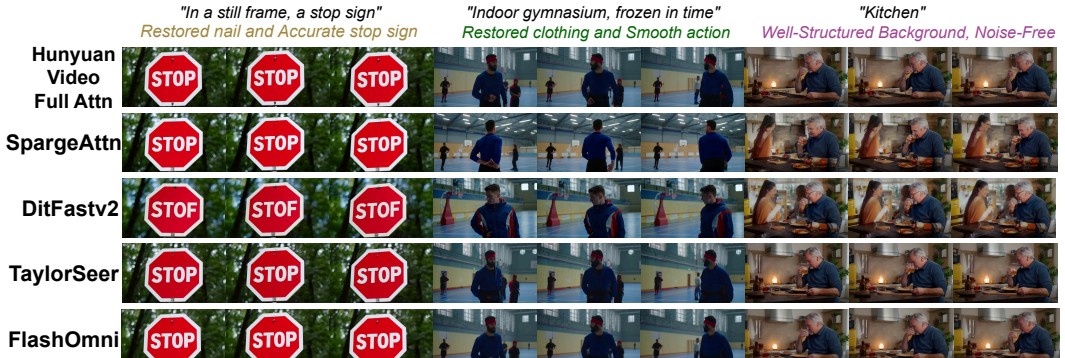

Figure 13: Visualization results for different acceleration methods on HunyuanVideo.

