# OpenReview forum: "FlashOmni: A Unified Sparse Attention Engine for Diffusion Transformers"
_ICLR.cc/2026/Conference — Submitted to ICLR 2026_

### Official Review · Reviewer_85zg · 2025-10-28

**Soundness:** 3
**Presentation:** 2
**Contribution:** 2
**Rating:** 2
**Confidence:** 4

**Summary:**

The paper proposes FlashOmni, a sparse attention for Diffusion Transformers (DiTs). It introduces sparse symbols, which are a unified abstraction that represents multiple sparsity strategies in a general sparse implementation. FlashOmni also proposes some methods to eliminate redundant computations. Experiments across text-to-image and text-to-video generation tasks demonstrate a high attention acceleration and up to 1.5× end-to-end speedup on large models without loss of visual quality.

**Strengths:**

The sparse symbols proposed in the paper are reasonable.

The paper provides an engineering contribution, including detailed kernel-level designs.

Although algorithmically limited, the proposed framework may serve as a useful system integration.

**Weaknesses:**

- **Motivation not well justified.** The paper claims that the current sparse attention design space is fragmented and prevents reuse across different applications. However, this claim is not convincing. Both static and dynamic sparsity methods are application-driven. Different methods are developed to address distinct modeling challenges rather than due to a lack of reusability. In practice, there already exist unified and highly usable APIs (e.g., *SpargeAttn*, *FlexAttn*) that support a customized sparse FlashAttention API, easily supporting both dynamic and static sparse attention implementation.

- **Limited originality of the proposed sparse symbols.** The introduced sparse symbols ($S_c$) and ($S_s$) are essentially a *mask encoding trick* rather than a new method. Representing sparse masks with 8-bit values only compresses storage by at most 8×, which is a naive bit-packing implementation. Compared with already widely used lookup-table (LUT) (such as in VSA or SpargeAttn), this approach provides negligible memory savings and no algorithmic novelty.

- **Implementation-level nature of GEMM-Q and GEMM-O.** The proposed GEMM-Q and GEMM-O kernels are primarily caching tricks. They do not fundamentally improve the theoretical efficiency of sparse attention. Their contribution is mostly engineering-level rather than algorithmic.

- **No complete end-to-end speedup experiments.** There is no complete end-to-end model speedup comparison experiments. Additionally, Figures 6 and 8 use the term *inference performance*, which is not clear. The paper should clarify that these are attention speed.

**Questions:**

**Questions about experimental results.** The reported performance of FlashAttention on A100 GPUs (93 TOPS) appears significantly lower than expected, as prior works and open benchmarks report values in the range of 140–190 TOPS. Similarly, *SpargeAttn*, which is based on *SageAttention*, should achieve well above 250 TOPS even without sparsity on HunyuanVideo. There may be possible benchmarking or configuration issues that need clarification.

---

> ### Author Response · Authors · 2025-11-19
>
> We thank the reviewer-85zg for the comments and address the major concerns below.
>
> -------------------
> **[Weaknesses-1: Motivation not well justified]**
>
> Reply:
> Indeed, current sparsity methods are largely application-driven, with different approaches addressing distinct challenges, whether static or dynamic. Existing libraries are usually built around specific designs, making it difficult for users to customize or compose arbitrary sparsity strategies using their existing kernels. Consequently, this requires significant amounts of time and effort for kernel design and implementation. This is what we call reusability.   **Our work does not lie on whether sparse attention is dynamic or static only, but rather in extending the support of the current sparse attention framework to varying sparsity granularities**.
>
> Existing sparse acceleration works generally fall into two separated categories:
> (1)  sparse attention kernels, whose kernel implements are limited to coarse-grained block skipping (e.g., SpargeAttention[11], DitFastAttnv2[12]), and (2) feature caching methods lack efficient kernel implements (e.g., ToCa[14] in token-wise feature reuse). **Without a general kernel, these two strategies cannot be applied simultaneously within one layer, which largely restricts the flexibility and composability of sparse strategy designs.**
>
> Beyond that, our work **not only supports different sparsity levels within a single attention computation**, allowing different Q-blocks (CTAs) to seamlessly adopt distinct sparsity granularities, **but also provides a higher kernel-level efficiency with less overhead, showing that our kernel design achieves lower runtime overhead while maintaining extensibility to different sparsity granularities** (please refer to the response **[Questions about experimental results.]** for detailed quantitative comparison). (common revision plan#4)
>
> --------------------
> **[Weaknesses-2: Limited originality of the proposed sparse symbols.]**
>
> Reply:
> **As we previously said, our work provides an unified framework allowing users to utilize not only block-skipping and feature-caching sparsity within a single transformer layer.** Under this scenario, a mask matrix is necessary for representing the sparse computation, while LUT-based methods suffer from significant memory consumption.
>
> Theoretically, let H denote the number of heads, $Size_f$ the symbol size for feature caching, $Size_b$ the symbol size for block skipping, N the number of tokens (Q/K/V), L the number of transformer blocks.
>
> (1). Memory overhead:
>
> Sparse mask:
>
> Our work consumes  $\mathcal{S}_s$: $  L \times (H \times \frac{N}{Size_f} \times \frac{N} {Size_b} )/ 8  $ B and $\mathcal{S}_c$: $ L \times  (H \times \frac{N}{Size_f}) / 8  $ B;
>
> while SpargeAttn consumes LUT:  $ L \times (H \times \frac{N}{Size_f} \times \frac{N}{Size_b}) × 4 $ B.
>
> And in practice, setting $Size_f$ =$ Size_b$ = 128 for the HunyunVideo model (Table 1, p.7) and generating videos with shape 544 × 960 × 61 (≈33 K tokens), our sparse symbols occupy about 11 MB, while SpargeAttn’s LUT consumes around 714 MB, **resulting in over 60× memory reduction**.
>
> Also, **utilizing mask matrices requires carefully designed kernels because of additional overheads on CUDA cores**. The originality comes from challenges and optimizations while using these mask matrices in the unified representation for both BS-level and FC-level sparsity patterns. **We need to balance the execution between CUDA cores and Tensor cores in order to improve efficiency and introduce less overheads**. We explicitly address this in **Subsection 3.4 page 5**, and the corresponding performance details please refer to following response **[Questions about experimental results.]**.  (common revision plan#2&#4)

---

> > ### Author Response · Authors · 2025-11-19
> >
> > ------------------
> > **[Weaknesses-3: Implementation-level nature of GEMM-Q and GEMM-O.]**
> >
> > Reply:
> > Thank you for your comments, but we believe there may be a misunderstanding. **The goal of GEMM-Q/O is not to improve the theoretical sparsity limit of attention, but rather to further optimize kernel efficiency once the sparsity has been determined**.
> >
> > In FlashOmni-Attention, **the actual sparsity is determined by user-defined sparsity strategies**, which in turn dictate the achievable sparsity level. Within this framework, **the optimization potential of GEMM stems from the feature-caching granularity**, a key factor contributing to the achievable sparsity level.
> >
> > **We respectfully disagree with the reviewer’s characterization of our GEMM-Q and GEMM-O kernels as mere “caching tricks.”** Existing sparse acceleration methods cannot accommodate multi-sparse strategies within a single layer, which limits the flexibility of sparsity designs and thus precludes any GEMM-level optimization. **In contrast, our FlashOmni framework expands sparsity granularity to the feature-caching level, unlocking new optimization opportunities for GEMM**. This capability is thoroughly analyzed in **Section 3.5 (page 6)**. Therefore, GEMM-Q and GEMM-O are not ”caching tricks“, but necessary kernel‑level optimizations uniquely enabled by FlashOmni’s extended sparse attention semantics. **While this  optimization  could not be realized by prior methods.**
> >
> > -------------------

---

> > > ### Comment · Reviewer_85zg · 2025-11-22
> > >
> > > Regarding the response to Weakness 3, I do not think there is a misunderstanding. I fully understand that *“the goal of GEMM-Q/O is not to improve the theoretical sparsity limit of attention, but rather to further optimize kernel efficiency once the sparsity has been determined.”*
> > >
> > > Therefore, the original weakness 3 is: *Implementation-level nature of GEMM-Q and GEMM-O. The proposed GEMM-Q and GEMM-O kernels are primarily caching optimizations. They do not fundamentally improve the theoretical efficiency of sparse attention; their contribution is mainly at the engineering level rather than at the algorithmic level.*
> > >
> > > I don’t think W3 reflects any misunderstanding you proposed.

---

> ### Author Response · Authors · 2025-11-19
>
> **[Weaknesses-4: end-to-end speedup experiments.]**
>
> Reply:
> Thank you for your valuable comment. Following your suggestion, we have added a new set of end-to-end performance comparisons on the Hunyuan model against existing sparse attention kernel libraries(SpargeAttn[11], DitFastAttnV2[12]) on A100. The updated results include both absolute inference latency and attention speedup ratios.  The result is shown as follows (detailed nvprof data refers to link[16]) and will be incorporated into our revised version (described in the common revision plan#1).
>
> | Method    |  Sparsity |End-to-end latency(s) | Attn latency(s)  | Preprocess latency(s) |
> |-----------|-----------|--------|--------|--------|
> | Diffusers-EfficientAttention | 0| 708  | 516.1 | 0 |
> | Diffusers-FlashAttention  |   0| 419  | 218| 0 |
> | SpargeAttn  | 32% | 401 | 171.5  | ~30 |
> | DitFastAttnV2 |  31% | 332 | 131  | 0 |
> | FlashOmni (50%,5%,6,1,30%)  | 47% | 278 | 109.1 | ~0.4|
>
> SpargeAttn uses a dynamic attention kernel but suffers from insufficient overhead optimization  (dequant and judge-op in attention) and heavy pre-processing costs (dynamic mask and quant overhead for kernel), leading to limited acceleration and reproducibility issues [1, 2]. In contrast, DitFastAttn is a static sparse attention mechanism that avoids extra overhead once the kernel is fixed, offering higher efficiency at equal sparsity but lower flexibility and generation quality (refer to Table 1 page 7).
>
> Our kernel **also supports dynamic attention and provides better overhead and preprocessing optimizations than SpargeAttn, achieving stronger performance** (detailed analysis please refer to response to **[Questions about experimental results.]**). Moreover, **it overcomes previous limitations of only supporting block skipping or feature caching within a single layer, enabling more flexible sparse strategy combinations for improved adaptability and generation quality.**
>
> In addition, we will clarify the meaning of “inference performance” in Figures 6 (page 8)and 8 (page 9)to explicitly indicate that it refers to single kernel performance. Detailed updates are described in the common revision plan#5.

---

> > ### Author Response · Authors · 2025-11-19
> >
> > **[Questions-1: Questions about experimental results.]**
> >
> > Reply:
> > (1) **Misunderstanding**. Thank you for your question. The reported $attn_{op}$ in our paper refers to the number of **fused multiply–add** (FMA) operations, which **differs from prior benchmarks that often count a multiply operator and an add operator as two operators**. This difference causes the lower values compared to 140–190 TOPS in prior works.
> >
> > In order to express more clearly, we define our count standard as follows:
> >
> > **TFOPS**:  Tera Fused Operations per Second. (original TOPS in paper)
> >
> > **TFOPs**:  Tera Fused Operations. (original TOPs in paper)
> >
> > **TOPS**:  Tera Operations per Second, which counts multiply and add into two operators.
> >
> > **TOPs**:  Tera Operations, which counts multiply and add into two operators.
> >
> > Our Flux model measurements of Table 1 (p. 7)  are for the generation of *1024×1024* images (4.5K tokens). The *TOPs* of $attn_{op}$ is formulated as follows:
> >
> > $
> > attn_{op} = 24 \times 4608 \times 4608 \times 128 \times 2 \times 2 \times 50 \times 57 / 10^{12} = 743.62\ \text{TOPs}
> > $
> >
> > We re-tested FullAttn, SpargeAttn, and FlashOmni using nvprof. Detailed NVIDIA profiling data are included in the supplementary link[7]. The summary results are as follows:
> >
> > | Method    | Sparsity | Attn_Latency(s) | TOPS   | TFOPS |
> > |-----------|----------|------------|--------|---------|
> > | FullAttn      | 0%        | 4.138      | 179.71 | 89.82   |
> > | SpargeAttn |  0% (without sparsity)  |   4.973  | 149.52 | 74.76 |
> > | SpargeAttn |  22%   | 3.832   |194.06 |  97.027 |
> > | FlashOmni | 28%      | 3.17       | 234.58 | 117.29  |
> > | FlashOmni | 46%      | 2.383      | 312.45 | 156.22  |
> >
> > The results are consistent with your expectations, and we will make a revision to our updated paper to use the same standard metrics.
> >
> > -------
> >
> > (2). Regarding the second point, “SpargeAttn, based on SageAttention, should achieve well above **250** TOPS even without sparsity on HunyuanVideo over an A100 GPU”,  we have conducted the experiment you suggested. **We can not reproduce the performance you mentioned. The detailed NVIDIA profiled data on end-to-end tests are included in the reference link[8]**.
> >
> > Besides the above results on Flux, our HunyuanVideo model measurements of Table 1 (p.7)  are for the generation of *544x960x61* videos (33K tokens). The *TOPs* of $attn_{op}$ is formulated as follows:
> >
> >
> > $
> > attn_{op} = 24 \times 32640 \times 32640 \times 128 \times 2 \times 2 \times 50 \times 60 / 10^{12} = 39273.7849344\ \text{TOPs}
> > $
> >
> > We re-tested FullAttn, SpargeAttn, and FlashOmni using nvprof. Detailed NVIDIA profiling data are included in the supplementary link[8]. The summary results are as follows:
> >
> > | Method    | Sparsity | Attn_Latency(s) | TOPS   | TFOPS |
> > |-----------|----------|------------|--------|---------|
> > | FullAttn      | 0%        | 217.882     |  180.253|   90.126 |
> > | SpargeAttn |    0% (without sparsity)   |    254.944 | 154.05  | 77.024  |
> > | SpargeAttn |   32%  | 180.466  | 217.624 | 108.81 |
> > | FlashOmni |     47%  |   114.6  | 342.7 | 171.35 |
> >
> >
> > (3). kernel performance:
> > We also evaluate the additional overhead within the designed Attention(e.g., decoding in FlashOmni, dequant and sparsity judgement in SpargeAttn) without any sparsity.
> > Our Flux model measurements of Table 1 (p. 7)  are for the generation of *1024×1024* images (4.5K tokens). The *Giga  OPs* of $attn_{op}$ is formulated as follows, detailed nvprof data refers to reference link[9]:
> >
> > $
> > attn_{op} = 24 \times 4608 \times 4608 \times 128 \times 2 \times 2 / 10^{9} = 260.92\ \text{GOPs}
> > $
> >
> > | Method    |  Attn_Latency(ms) | TOPS   | TFOPS |
> > |-----------|-----------|--------|---------|
> > | Scale_dot     |   1.364    | 191.29 |   95.64 |
> > | FlashInfer     | 1.253    | 208.24 | 104.12   |
> > | SpargeAttn |   1.615 | 161.56 |80.78 |
> > | FlashOmni  | 1.362  |  191.57  | 95.79 |
> >
> > Our HunyuanVideo model measurements of Table 1 (p.7)  are for the generation of *544x960x61* videos (33K tokens).  The *Tera OPs* of $attn_{op}$ is formulated as follows, detailed nvprof data refers to the reference link[10]:
> >
> > $
> > attn_{op} = 24 \times 32640 \times 32640 \times 128 \times 2 \times 2 / 10^{12} = 13.09\ \text{TOPs}
> > $
> >
> > | Method    |  Attn_Latency(ms) | TOPS   | TFOPS |
> > |-----------|-----------|--------|---------|
> > | Scale_dot   |   71.98   | 181.87 |   90.94 |
> > | FlashInfer   | 66.765   | 196.08 | 98.04   |
> > | SpargeAttn |   80.02 | 163.58 | 81.79 |
> > | FlashOmni  |  71.72| 182.53  | 91.27 |
> >
> > **Experimental results show that without any sparsity, FlashOmni achieves better kernel optimization, indicating its superior ability to reduce the extra overhead caused by CUDA core operations compared with SpargeAttn. And further extend support at the feature caching level.**
> >
> > **Furthermore, we checked the SpargeAttention GitHub repository and found similar reports from other users citing difficulty reproducing the expected performance both on A100 and H100 hardware, as referenced in links [1,2]**.
> >
> >
> > ------------------------

---

> > > ### Author Response · Authors · 2025-11-19
> > >
> > > (4). pre-process overhead for dynamic attention kernel:
> > >
> > > In SpargeAttn, the per-iteration generation of the mask according to tuned thresholds, combined with QK quantization, brings a **huge pre-process overhead**. **Especially, SpargeAttnprocess adds 30 seconds of overhead in HunyuanVideo and 4 seconds of overhead in Flux** (see triton_bmm_pool_sim_simmean_fuse_quant item in link[1,2]). **Notably, even in the Flux scenario, this additional overhead is almost close to the time cost of full attention computation itself**. In contrast, in our method, the Triton kernel for generating sparse symbols and the compression kernel contribute **negligible runtime overhead** (see triton_fill_flashomni_kernel and flashomni::quantization::SegmentPackBitsKernel in the referenced link[1,2]).
> > >
> > >
> > > (5). Moreover, our approach is **tuning‑free**, incurring no pre‑tuning overhead. In contrast, **SpargeAttn requires significant GPU resources for tuning**, as referenced in links [3,4,5,6]. Even after extensive tuning, its performance remains inferior to ours, both in generation quality and kernel efficiency (please refer to Tables 1 and 2 on page 7).Without tuning, it is hardly usable in practice.
> > >
> > > -------------------
> > > In conclusion, there is confusion about the definition of the TOPs metric, and we will clarify it in the revised version. Besides, we have thoroughly validated that **FlashOmni consistently outperforms SparseAttn** in terms of memory overhead, pre‑processing cost, kernel efficiency, end‑to‑end sparsity, and generation quality, as well as hyperparameter tuning cost, with strong evidence from nvprof profiling results. Our work **not only supports different sparsity levels within a single attention computation**, allowing different Q-blocks (CTAs) to seamlessly adopt distinct sparsity granularities, **but also provides a higher kernel-level efficiency with less overhead, showing that our kernel design achieves lower runtime overhead while maintaining extensibility to different sparsity granularities.**
> > >
> > > Detailed updates are described in the common revision plan#2&#4.
> > >
> > > ------------------------------
> > >
> > > **Reference**
> > >
> > > [1].https://github.com/thu-ml/SpargeAttn/issues/18
> > >
> > > [2].https://github.com/thu-ml/SpargeAttn/issues/67
> > >
> > > [3].https://github.com/thu-ml/SpargeAttn/issues/50
> > >
> > > [4].https://github.com/thu-ml/SpargeAttn/issues/6
> > >
> > > [5].https://github.com/thu-ml/SpargeAttn/issues/53
> > >
> > > [6].https://github.com/thu-ml/SpargeAttn/issues/15
> > >
> > > [7]./FlashOmni/example/flux/nvprof
> > >
> > > [8]./FlashOmni/example/hunyuan/nvprof
> > >
> > > [9]./FlashOmni/benchmark/nvprof_attn/flux
> > >
> > > [10]./FlashOmni/benchmark/nvprof_attn/hunyuan
> > >
> > > [11]. Zhang, J., Xiang, C., Huang, H., Xi, H., Zhu, J., & Chen, J. SpargeAttention: Accurate and Training-free Sparse Attention Accelerating Any Model Inference. ICML25.
> > >
> > > [12]. Zhang, H., Su, R., Yuan, Z., Chen, P., Shen, M., Fan, Y., ... & Wang, Y. DiTFastAttnV2: Head-wise Attention Compression for Multi-Modality Diffusion Transformers. ICCV25.
> > >
> > > [13]. Liu, J., Zou, C., Lyu, Y., Chen, J., & Zhang, L. From reusing to forecasting: Accelerating diffusion models with taylorseers.ICCV25.
> > >
> > > [14]. Zou, C., Liu, X., Liu, T., Huang, S., & Zhang, L. Accelerating Diffusion Transformers with Token-wise Feature Caching. ICLR25.
> > >
> > > [15]. Dao, T. FlashAttention-2: Faster Attention with Better Parallelism and Work Partitioning. ICLR24.
> > >
> > > [16]. /FlashOmni/example/hunyuan/nvprof/e2e

---

> ### Comment · Reviewer_85zg · 2025-11-20
> **Fix TOPS and compare speed on more GPUs**
>
> Thank you for the detailed response.
>
> 1. **Correction of TOPS in the paper**
>
>    From your explanation, it is clear that the TOPS reported in the original version of your paper is not using the standard TOPS definition. In the revision, please update all reported TOPS numbers to use the correct TOPS definition.
>
> 2. **Need speed comparison on more GPUs**
>
>    For a method that aims to be generally useful, it is important to demonstrate speedups on different GPUs, not only on A100. In addition, in practice, diffusion models are often deployed on GPUs used for inference, such as RTX 4090 and RTX 5090. Therefore, the paper should also report speed comparisons on **RTX 4090**, **RTX 5090**, **H100**, etc. By the way, for SpargeAttn, you can directly benchmark using the `spas_sage2_attn_meansim_topk_cuda` API.
>
> Additionally, some diffusion models do not use `headdim=128`, attention speed comparison on `headdim=64` is also needed.

---

> > ### Author Response · Authors · 2025-11-24
> >
> > We thank the reviewer-85zg for the additional comments and address the major concerns below.
> >
> > **[Suggestion-1: Correction of TOPS in the paper.]**
> >
> > Reply:
> > Thank you for your valuable suggestion. We appreciate your careful feedback, and we will revise our paper accordingly to correct the metric definition and update all reported TOPS numbers in the revision.

---

> ### Comment · Reviewer_85zg · 2025-11-20
>
> Thank you again for the additional analysis.
>
> SpargeAttn also seems to support a *tuning-free usage*. In this mode, it can be used directly for inference without any offline tuning or threshold search.
>
> Could you also report end-to-end video generation quality when both methods are used in a tuning-free configuration for fairness? For SpargeAttn, this can be done by directly using APIs of `spas_sage_attn_meansim_topk_cuda`, or `spas_sage2_attn_meansim_topk_cuda` for inference, without running any tuning procedure.

---

> > ### Author Response · Authors · 2025-11-24
> >
> > **[Suggestion-2: Tuning-free comparsion.]**
> >
> > Reply:
> > Tuning is an essential optimization search procedure in SpargeAttn, designed to find the best hyperparameters for maximizing video generation quality. In contrast, FlashOmni uses fixed hyperparameters, which are not necessarily the optimal settings for peak video quality. Moreover, in our end‑to‑end speed measurements, **we did not include the tuning time for SpargeAttn, ensuring a fair comparison**.
> >
> > Additionally, when using the spas_sage_attn_meansim_topk_cuda API in HunyuanVideo[7,8], the SpargeAttn method degenerates into Minerence[3] + Sage[2], **losing its most important innovation** ( the tuning-based selection for $QK^T$  and  $PV$ sparsity). Nevertheless, to make the comparison more convincing, we performed an extra experiment under untuned configuration as your suggestion.
> >
> > Due to time constraints, we were able to complete only the first‑fold evaluation at this stage. While full vBench validation typically requires multiple folds, we first conducted a similarity‑based analysis on the first fold and will include the complete comparison in the revision.
> >
> > We compared both methods in a tuning‑free configuration. Specifically, we evaluated untuned SpargeAttn at a lower sparsity level (27%) using the spas_sage_attn_meansim_topk_cuda API, and untuned FlashOmni at significantly higher avg sparsity levels (47% and 39%). The detailed sparsity density please refer to Figure.7-page 9.
> >
> > **SpargeAttn(untuned) with lower sparsity (27%) still performs worse across all quality metrics than FlashOmni with much higher sparsity (47% and 39%). This demonstrates the inherent efficiency and quality of FlashOmni**. The results are summarized below:
> >
> > The detailed results are shown as follows:
> > | Method    | Sparsity| PSNR | LPIPS   | SSIM |
> > |-----------|-----------|--------|---------|---------|
> > | Sparge(tuned)  | 32%|21.4619   |     0.2381      |       0.7844      |
> > | Sparge(no tuned)  | 27%| 23.8222  | 0.1914 | 0.813  |
> > | FlashOmni (50%,5%,6,1,30%)  | 47%|  27.6576   |0.1185|  0.8733 |
> > | FlashOmni (40%,1%,6,1,0) |39%|  28.5233  | 0.0921 |  0.8958  |

---

> ### Comment · Reviewer_85zg · 2025-11-22
>
> Thank you for adding the end-to-end speed experiment.
> For a method that aims to be generally useful, it is important to demonstrate speedups on different GPUs, not only on A100. Moreover, in practice, diffusion models are often deployed on GPUs used for inference, such as RTX 4090 and RTX 5090.
> Therefore, I suggest also reporting kernel and end-to-end speed comparisons on RTX 4090, RTX 5090, H100, etc.

---

> > ### Author Response · Authors · 2025-11-24
> >
> > **[Suggestion-3: Speed comparison on more GPUs]**
> >
> > Reply:
> > We thank the reviewer for the constructive feedback. Following your suggestion, we conducted additional kernel performance comparisons on an RTX 4090 **under matched sparsity levels (since spargeattn top‑k api cannot express the feature‑caching‑level sparsity used in FlashOmni)** for different head dimensions (e.g., 64 and 128).  The detailed nvprof profiling data are provided in the supplementary material (see Ref. [1]). The results are summarized in the table below(Sparsity-#tokens_#head_dim).
> >
> > To ensure a fair comparison with SpargeAttn, we implemented an 8‑bit kernel based on SageAttention[2] and compared it with the recommended API (spas_sage2_attn_meansim_topk_cuda). This API[7] essentially corresponds to MInference[3] + SageAttention[2] and does not incorporate the sparsity algorithmic innovations (Hyper-parameter tuning) proposed in the SpargeAttn paper. **Under the same sparsity settings, our method achieves comparable computational efficiency, indicating that our design does not sacrifice compute performance.**
> >
> > | (Sparsity-4608_128)  | SageAttention | SpargeAttention_topk  | FlashOmni-8bit   |
> > |-----------|-----------|--------|--------|
> > | 14%   | 0.677  |  0.613  | 0.551 |
> > | 20%   | - | 0.573  | 0.55 |
> > | 37%   | - | 0.461  | 0.483 |
> > | 44%   | - | 0.419  | 0.409 |
> > | 55%   | - | 0.339  | 0.342 |
> > | 60%   | - | 0.300  | 0.305 |
> >
> > | (Sparsity-16896_128)  | SageAttention | SpargeAttention_topk  | FlashOmni-8bit   |
> > |-----------|-----------|--------|--------|
> > | 14%   | 8.505  |  7.621  | 7.302  |
> > | 28%   | - | 6.396  | 5.985  |
> > | 37%   | - | 5.594  | 5.501 |
> > | 44%   | - | 4.960  | 4.725 |
> > | 52%   | - | 4.26   | 4.235 |
> > | 64%   | - | 3.206  | 3.209 |
> > | 76%   | - | 2.141  | 2.087 |
> >
> > | (Sparsity-16896_64)  | SageAttention | SpargeAttention_topk  | FlashOmni-8bit   |
> > |-----------|-----------|--------|--------|
> > | 14%   | 4.578  | 4.480  | 4.554  |
> > | 28%   | - | 3.835  | 3.772  |
> > | 37%   | - | 3.354  | 3.387 |
> > | 44%   | - | 2.974  | 2.986 |
> > | 52%   | - | 2.553  | 2.559 |
> > | 64%   | - | 1.932  | 1.963 |
> >
> > **And crucially, our method aggressively optimizes the memory of sparse symbols**: our sparse symbols occupy only about 11 MB, whereas SpargeAttn’s LUT requires around 714 MB, yielding more than a 60× reduction in additional memory overhead (we mentioned before). **At the same time, our sparse-symbol design enables a single kernel to jointly support both feature caching and block‑skipping sparsity strategies. In contrast, the provided API (top‑k or CDF in spargeattn) only supports block‑skipping level, without unified support for feature caching within a single layer.**
> >
> > For the end‑to‑end experiments, we have demonstrated that at higher sparsity levels, FlashOmni still delivers significantly better video quality than Sparge(both tuning and untuned), please refer to the above reply. On RTX 4090, the attention kernels achieve comparable performance to Sparge at the same sparsity level, and FlashOmni further benefits from GEMM computation optimizations. Therefore, under the same sparsity configuration used in the above reply, FlashOmni with higher sparsity is theoretically expected to achieve higher end‑to‑end inference performance on the 4090.

---

> ### Author Response · Authors · 2025-11-24
>
> **[Weakness-1:Implementation-level nature of GEMM-Q and GEMM-O.]**
>
> Reply:
> Thank you for the feedback. **Trading memory for computation is a standard and well‑accepted technique in modern sequence models. A typical example is KV Cache[6] in LLMs**: by storing past key/value, one avoids recomputing linear for previous tokens, so only the latest token requires a full linear layer. This is a core mechanism for efficient autoregressive decoding.
>
> **Our diffusion setting is more challenging than token‑by‑token generation in LLMs**. With FlashOmni‑attention’s support for token‑wise feature caching, a subset of tokens across timesteps can reuse their previous attention outputs, so the corresponding linear (GEMM) computations can be skipped. **Unlike LLMs, where the reused tokens always form a fixed prefix pattern. In diffusion attention,  the feature-caching tokens are dynamic, potentially non‑contiguous, and induce sparsity along multiple dimensions in the corresponding linear**. This calls for GEMM kernels with fine‑grained per‑CTA skipping logic along different dimensions, rather than the simple “skip all previous N−1 tokens” pattern used in LLM KV caching.
>
> **GEMM‑Q/O are designed precisely for these more complex sparsity patterns. “Caching” is just the visible tip of the underlying algorithmic design**: how we encode sparsity, map it to GEMM tiles, and coordinate skipping across dimensions. And crucially, it is **FlashOmni’s design that makes this type of GEMM optimization possible; other sota sparse attention[4,5] don't have the chance** to optimize it due to lacking support for token-wise feature-caching within attention. **If one were to label all such memory‑for‑compute designs as “just caching tricks”, then KV Cache in LLMs would fall into the same category, despite being widely recognized as a key algorithmic component for efficient inference.**
>
> Moreover, in FlashOmni the kernels are executed in the order GEMM-Q --> Attention --> GEMM-O. For a given sparsity combination pattern, each kernel can eliminate its corresponding redundant computations, and the **caching is only incurred in the final GEMM-O; the other two kernels introduce no extra memory overhead**.
>
> **From an AI infrastructure perspective, the most direct and impactful way to improve inference efficiency is to better exploit existing algorithmic sparsity through system and kernel design**. Our contributions sit exactly at this algorithm–system interface, and the GEMM optimization we propose is only one component of the overall FlashOmni framework. While GEMM‑Q/O do not change the theoretical sparsity limit of attention, they provide new sparse‑symbol‑guided GEMM kernel designs that  **(i) support multi‑dimensional sparsity within GEMM and (ii) turn feature‑caching‑induced sparsity in DiTs into real savings in linear computation, analogous to how KV Cache improves inference efficiency in LLMs.**
>
>
> -------------------------
>
>
> **Reference:**
>
> [1]./FlashOmni/benchmark/nvprof_attn/4090
>
> [2].Zhang, J., Wei, J., Huang, H., Zhang, P., Zhu, J., & Chen, J. Sageattention: Accurate 8-bit attention for plug-and-play inference acceleration.ICLR25.
>
> [3].Jiang, H., Li, Y., Zhang, C., Wu, Q., Luo, X., Ahn, S., ... & Qiu, L. Minference 1.0: Accelerating pre-filling for long-context llms via dynamic sparse attention. NeurIPS24.
>
> [4]. Zhang, J., Xiang, C., Huang, H., Xi, H., Zhu, J., & Chen, J. SpargeAttention: Accurate and Training-free Sparse Attention Accelerating Any Model Inference. ICML25.
>
> [5]. Zhang, H., Su, R., Yuan, Z., Chen, P., Shen, M., Fan, Y., ... & Wang, Y. DiTFastAttnV2: Head-wise Attention Compression for Multi-Modality Diffusion Transformers. ICCV25.
>
> [6].https://github.com/huggingface/transformers/blob/v4.57.2/src/transformers/generation/configuration_utils.py#L138
>
> [7].https://github.com/thu-ml/SpargeAttn/blob/main/spas_sage_attn/core.py#L100
>
> [8].https://github.com/thu-ml/SpargeAttn/blob/main/inference_examples/modify_model/modify_hunyuan.py#L108

---

> ### Author Response · Authors · 2025-11-27
>
> Dear reviewer 85zg,
>
> We sincerely apologize for disturbing you, and we truly appreciate the time you have already devoted to reviewing our paper. As the discussion period is approaching its end (with less than a week remaining), we have noticed that the score has been raised, for which we are very grateful. At the same time, we would like to confirm whether our responses have fully addressed all of your concerns. If there are any remaining issues or questions that you feel have not been adequately clarified, please do not hesitate to let us know. We will do our best to respond as quickly and thoroughly as possible.
>
> Thank you again for your time.

---

### Official Review · Reviewer_pugu · 2025-10-29

**Soundness:** 2
**Presentation:** 2
**Contribution:** 2
**Rating:** 2
**Confidence:** 3

**Summary:**

This paper proposes **FlashOmni**, a unified sparse attention engine aiming to overcome the fragmentation of existing sparse acceleration methods for Diffusion Transformers (DiTs) by integrating them into a single framework. The method introduces unified sparse symbols and optimized kernels to support diverse sparsity strategies.

**Strengths:**

* **Clarity and Visual Appeal**: The paper's figures and main body are well-presented and contribute to clarity in explaining the overall system design.

**Weaknesses:**

- **Incremental Novelty**: The primary weakness is the limited conceptual novelty. The paper's sparsity-finding logic is not new; it explicitly adopts TaylorSeer [1] for its feature caching strategy and aligns its block-skipping approach with SpargeAttention [2]. The contribution is therefore the combination of these two existing methods, which feels incremental.

- **Contribution Feels Limited to Implementation**: The main novelty—the unified kernel and sparse symbols—can be viewed as a clever but small, implementation-level engineering optimization rather than a new scientific algorithm. It is not clear that this systems-level integration, by itself, meets the high bar for ICLR.

- **Incremental Performance Gains and Unfair Comparison**: The reported end-to-end speedups (e.g., "~ 1.5x end-to-end acceleration" on Hunyuan) are positive but not transformative. While Table 1 and Table 2 show that FlashOmni achieves better quality metrics (like PSNR/LPIPS) than the individual baselines (SpargeAttn [2], TaylorSeer [1]), the gain from this complex "unification" is not clearly isolated. Thus the comparison is unfair.

[1] Liu, J., Zou, C., Lyu, Y., Chen, J., & Zhang, L. From reusing to forecasting: Accelerating diffusion models with taylorseers. In ICCV 2025.

[2] Zhang, J., Xiang, C., Huang, H., Xi, H., Zhu, J., & Chen, J. SpargeAttention: Accurate and Training-free Sparse Attention Accelerating Any Model Inference. In ICML 2025.

**Questions:**

- **Missing Baseline Data in Table 1 2**: In Table 1 2, the "Full-Attention" baseline for the "FLUX.1 [dev]: 50 steps" configuration is missing all quality metrics. The cells for PSNR, LPIPS, SSIM, CLIP-IQA, and FID are all blank. This is the most critical baseline for that experiment. Without these numbers, it is impossible to assess the quality degradation or preservation of any of the sparse methods, including the author's own. Can the authors please provide these crucial baseline numbers?

---

> ### Author Response · Authors · 2025-11-19
>
> We thank the reviewer-pugu for the comments and address the major concerns below.
>
> --------------
> **[Weaknesses-1: Incremental Novelty]**
>
> Reply:
> We appreciate the reviewer’s concern and we believe there must be a **misunderstanding on our core contribution**. Our **contribution does not lie in proposing new sparsity strategies, but rather in developing a more complete sparse attention framework that supports varying sparsity granularities within a single kernel**.
>
> Existing sparse acceleration works generally fall into two separated categories:
> (1)  sparse attention kernels, whose kernel implements are limited to coarse-grained block skipping (e.g., SpargeAttention[1], DitFastAttnv2[2]), and (2) feature caching methods lack efficient kernel implements (e.g., ToCa[4] in token-wise feature reuse).
> **Without a general kernel, these two strategies cannot be applied simultaneously within one layer, which largely restricts the flexibility and composability of sparse strategy designs**.
>
>
> Beyond that, our work **not only supports different sparsity levels within a single attention computation**, allowing different Q-blocks (CTAs) to seamlessly adopt distinct sparsity granularities, **but also provides a higher kernel-level efficiency with less overhead**, showing that our kernel design **achieves lower runtime overhead while maintaining extensibility to different sparsity granularities** (please refer to the response for Weaknesses2: **[Contribution Feels Limited to Implementation]** for detailed quantitative comparison).
>
>
> Regarding the sparsity strategies employed in our experiments, it was chosen only to verify the correctness and performance of our kernel during end-to-end inference. These strategies can be flexibly customized by users and are not the core designs we focus on, which do not represent a mere combination of prior methods. (Common revision plan#4)
>
> ------------------
> **[Weaknesses2:Contribution Feels Limited to Implementation]**
>
> Reply:
> We acknowledge the concern that our contribution may seem like a minor engineering optimization. We would like to clarify that our work focuses on the kernel-level optimization, that builds a kernel framework that enables flexible and efficient sparse attention across varying granularities.  To address this concern, we will present evidence from several aspects: (1).memory overhead, (2).pre-processing overhead, (3).kernel performance, (4)end-to-end sparsity and generation quality, and (5)hyperparameter tuning cost.
>
>
> (1). Memory overhead:
>
> Sparse mask:
>
> Our work consumes  $\mathcal{S}_s$: $  L \times (H \times \frac{N}{Size_f} \times \frac{N} {Size_b} )/ 8  $ B and $\mathcal{S}_c$: $ L \times  (H \times \frac{N}{Size_f}) / 8  $ B;
>
> while SpargeAttn consumes LUT:  $ L \times (H \times \frac{N}{Size_f} \times \frac{N}{Size_b}) × 4 $ B.
>
> And in practice, setting $Size_f$ =$ Size_b$ = 128 for the HunyunVideo model (Table 1, p.7) and generating videos with shape 544 × 960 × 61 (≈33 K tokens), our sparse symbols occupy about 11 MB, while SpargeAttn’s LUT consumes around 714 MB, **resulting in over 60× memory reduction**.
>
>
> Cache Bias:
>
> if did’t activate the feature caching, without cost, same with block skipping methods;
> When activating the feature caching strategy, the cache bias is used for storing the output of a transformer block(with  using GEMM-O) or attention output (without using GEMM-O), and it has the same size with feature caching-base methods.
>
> (2). pre-process overhead for dynamic attention kernel:
>
> In SpargeAttn, the per-iteration generation of the mask according to tuned thresholds, combined with QK quantization, brings a **pre-process overhead**. In this case, **this process adds 30 seconds of overhead in HunyuanVideo and 4 seconds of overhead in Flux** (see triton_bmm_pool_sim_simmean_fuse_quant item in link[6,7]). Notably, even in the Flux scenario, this additional overhead is almost close to the time cost of full attention computation itself. In contrast, in our method, the Triton kernel for generating sparse symbols and the compression kernel contribute **negligible runtime overhead** (see triton_fill_flashomni_kernel and flashomni::quantization::SegmentPackBitsKernel in the referenced link[6,7]).

---

> > ### Author Response · Authors · 2025-11-19
> >
> > (3). kernel performance:
> >
> > We also evaluate the additional overhead within a single Attention(e.g. decoding in FlashOmni, dequant and sparsity judgement in SpargeAttn) without any sparsity.
> > Our Flux model measurements of Table 1 (p. 7)  are for the generation of *1024×1024* images (4.5K tokens). The *Giga  OPs* of $attn_{op}$ is formulated as follows, detailed nvprof data refers to reference link[8]:
> >
> > $
> > attn_{op} = 24 \times 4608 \times 4608 \times 128 \times 2 \times 2 / 10^{9} = 260.92\ \text{GOPs}
> > $
> >
> > | Method    |  Attn_Latency(ms) | TOPS   | TFOPS |
> > |-----------|-----------|--------|---------|
> > | Scale_dot     |   1.364    | 191.29 |   95.64 |
> > | FlashInfer     | 1.253    | 208.24 | 104.12   |
> > | SpargeAttn |   1.615 | 161.56 |80.78 |
> > | FlashOmni  | 1.362  |  191.57  | 95.79 |
> >
> > Our HunyuanVideo model measurements of Table 1 (p.7)  are for the generation of *544x960x61* videos (33K tokens).  The *Tera OPs* of $attn_{op}$ is formulated as follows, detailed nvprof data refers to reference link[9]:
> >
> > $
> > attn_{op} = 24 \times 32640 \times 32640 \times 128 \times 2 \times 2 / 10^{12} = 13.09\ \text{TOPs}
> > $
> >
> > | Method    |  Attn_Latency(ms) | TOPS   | TFOPS |
> > |-----------|-----------|--------|---------|
> > | Scale_dot   |   71.98   | 181.87 |   90.94 |
> > | FlashInfer   | 66.765   | 196.08 | 98.04   |
> > | SpargeAttn |   80.02 | 163.58 | 81.79 |
> > | FlashOmni  |  71.72| 182.53  | 91.27 |
> >
> > Experimental results show that **without any sparsity, FlashOmni achieves better kernel optimization, indicating its superior ability to reduce the extra overhead caused by CUDA core operations compared with SpargeAttn. And further extend support at the feature caching level.**
> >
> > (4). End-to-end sparsity and generation quality, please refer to Table 1 (p. 7). The detailed reproduced data, please refer to the response **[Questions about experimental results.]**  to reviewer-85zg.
> >
> > (5). Tuning cost: Moreover, our approach is **tuning‑free**, incurring no pre‑tuning overhead. In contrast, SpargeAttn requires significant GPU resources for tuning, as referenced in links [12,13,14,15].
> >
> > **These results show that our unified kernel design offers fundamental and generalizable systems-level advances beyond a simple implementation tweak**.(Common revision plan#2&#4)
> >
> > ---------------
> > **[Weaknesses-3: Incremental Performance Gains and Unfair Comparison]**
> >
> > Reply:
> > Thank you for raising this concern.  Regarding the "not transformative",we add a new set of end-to-end performance comparisons on the Hunyuan model against existing sparse attention kernel libraries(SpargeAttn[1], DitFastAttnV2[2]) on A100. The result is shown as follows (detailed nvprof data refers to link[16]) (described in the common revision plan#1).
> >
> > | Method    |  Sparsity |End-to-end latency(s) | Attn latency(s)  | Preprocess latency(s) |
> > |-----------|-----------|--------|--------|--------|
> > | Diffusers-EfficientAttention | 0| 708  | 516.1 | 0 |
> > | Diffusers-FlashAttention  |   0| 419  | 218| 0 |
> > | SpargeAttn  | 32% | 401 | 171.5  | ~30 |
> > | DitFastAttnV2 |  31% | 332 | 131  | 0 |
> > | FlashOmni (50%,5%,6,1,30%)  | 47% | 278 | 109.1 | ~0.4|
> >
> > SpargeAttn uses a dynamic attention kernel but suffers from insufficient overhead optimization  (dequant and judge-op in attention) and heavy pre-processing costs (dynamic mask and quant overhead for kernel), leading to limited acceleration and reproducibility issues [10, 11]. In contrast, DitFastAttn is a static sparse attention mechanism that avoids extra overhead once the kernel is fixed, offering higher efficiency at equal sparsity but lower flexibility and generation quality (refer to Table 1 page 7).
> >
> > Our kernel **also supports dynamic attention** and **provides better overhead and preprocessing optimizations than SpargeAttn**, achieving stronger performance (detailed analysis please refer to next response **[Questions about experimental results.]** to reviewer-85zg). **Moreover, it overcomes previous limitations of only supporting using block skipping or feature caching within a single layer, enabling more flexible sparse strategy combinations for improved adaptability and generation quality**.
> >
> > Regarding the concern about **unfair comparison**, we would like to clarify that among all compared methods, **only our attention kernel supports simultaneous use of multiple sparsity granularities within the same layer, whereas other baselines do not. Therefore, the comparison is inherently fair since our unified design enables capabilities unavailable in prior implementations**.
> >
> > Regarding the concern about “complex unification,” our kernel provides a systematic and extensible framework that enables users to flexibly define and combine diverse sparsity strategies, **going beyond the limited patterns supported by prior methods. This design ensures both adaptability and simplicity while achieving consistently superior performance than recent SOTA implementations (e.g., SpargeAttn) across multiple dimensions** (refer to above response).

---

> > > ### Author Response · Authors · 2025-11-19
> > >
> > > **[Questions-1: Missing Baseline Data in Table 1 2]**
> > >
> > > Reply:
> > > Thank you for your question and for carefully reviewing our tables. We believe there may have been a misunderstanding when reading Tables 1 and 2.
> > >
> > > (1)The **CLIP-IQA** value for the “Full-Attention configuration” is **indeed** present in tables 1 and 2.
> > >
> > > (2)In our visual generation evaluation, sparse attention outputs are compared against Full-Attention results, which serve as the gold standard. **Similarity-based metrics** (PSNR, LPIPS, SSIM, FID) thus **measure the difference of output between every sparse method and Full-Attention; for the baseline itself, these scores would be trivial self-similarity and are therefore omitted intentionally**. This convention is consistent with prior work (e.g., DitFastAttnV2 [2]), where similarity metrics are only reported for methods being compared against the Full-Attention baseline rather than for the baseline itself.
> > >
> > > --------------------
> > >
> > > **Reference:**
> > >
> > > [1]. Zhang, J., Xiang, C., Huang, H., Xi, H., Zhu, J., & Chen, J. SpargeAttention: Accurate and Training-free Sparse Attention Accelerating Any Model Inference. ICML25.
> > >
> > > [2]. Zhang, H., Su, R., Yuan, Z., Chen, P., Shen, M., Fan, Y., ... & Wang, Y. DiTFastAttnV2: Head-wise Attention Compression for Multi-Modality Diffusion Transformers. ICCV25.
> > >
> > > [3]. Liu, J., Zou, C., Lyu, Y., Chen, J., & Zhang, L. From reusing to forecasting: Accelerating diffusion models with taylorseers.ICCV25.
> > >
> > > [4]. Zou, C., Liu, X., Liu, T., Huang, S., & Zhang, L. Accelerating Diffusion Transformers with Token-wise Feature Caching. ICLR25.
> > >
> > > [5]. Dao, T. FlashAttention-2: Faster Attention with Better Parallelism and Work Partitioning. ICLR24.
> > >
> > > [6]./FlashOmni/example/flux/nvprof
> > >
> > > [7]./FlashOmni/example/hunyuan/nvprof
> > >
> > > [8]./FlashOmni/benchmark/nvprof_attn/flux
> > >
> > > [9]./FlashOmni/benchmark/nvprof_attn/hunyuan
> > >
> > > [10].https://github.com/thu-ml/SpargeAttn/issues/18
> > >
> > > [11].https://github.com/thu-ml/SpargeAttn/issues/67
> > >
> > > [12].https://github.com/thu-ml/SpargeAttn/issues/50
> > >
> > > [13].https://github.com/thu-ml/SpargeAttn/issues/6
> > >
> > > [14].https://github.com/thu-ml/SpargeAttn/issues/53
> > >
> > > [15].https://github.com/thu-ml/SpargeAttn/issues/15
> > >
> > > [16]./FlashOmni/example/hunyuan/nvprof/e2e

---

> > > > ### Comment · Reviewer_pugu · 2025-11-27
> > > >
> > > > Thank you for your detailed response. I appreciate your effort. Most of my concerns are addressed. The acceleration results are good, and other parts of the answer clarify the ambiguity. However, I still feel most of the novelty heavily lies in the engineering part. Thus, I will increase my score to 4.

---

> > > > > ### Author Response · Authors · 2025-11-27
> > > > >
> > > > > Dear Reviewer pugu,
> > > > >
> > > > > Thank you for raising your score. Regarding your remark that “most of the novelty heavily lies in the engineering part,” we would like to clarify once again.
> > > > >
> > > > > **AI infrastructure and Library** (the primary area session to which we submitted)  **are inherently intertwined with system engineering optimization**, and FlashOmni focuses on the kernel optimization. In our view, **the most direct criteria for evaluating an operator library are its generality and its inference performance.**
> > > > >
> > > > > **FlashOmni provides a complete API stack that satisfies both**: **for generality**, the sparse‑symbols abstraction allows users to flexibly express and compose arbitrarily fine‑grained sparsity strategies, making it straightforward to implement custom sparsity patterns without touching low‑level kernels; **for performance**, our kernels are designed to eliminate all redundant computation in GEMM and attention,  introduced by sparsity, achieving substantial speedups while incurring almost no additional memory overhead.
> > > > >
> > > > > **Regarding the attention optimization, please refer to our earlier response.** Regarding the **GEMM optimization**, with support for token‑wise feature caching, a subset of tokens across timesteps can reuse their previous attention outputs. The core idea is **eliminating the redundant computation in linear, similar to  KVCache in LLM.** However, **this optimization for  DiT faces more challenges than LLMs**, where the reused tokens always form a fixed prefix pattern. In contrast, **the reused tokens are dynamic, potentially non‑contiguous, and induce multi‑dimensional sparsity in the linears**.  This calls for GEMM kernels with fine‑grained per‑CTA skipping logic along different dimensions, rather than the simple “skip all previous N−1 tokens” pattern used in LLM KV caching.
> > > > >
> > > > > Moreover, in FlashOmni the kernels are executed in the order GEMM-Q → Attention → GEMM-O. For a given sparsity combination pattern, **each kernel can eliminate its corresponding redundant computations, and caching is only incurred in the final GEMM-O; the other two kernels introduce no extra memory overhead**.
> > > > >
> > > > >
> > > > > From an **AI infrastructure and library perspective**, the most direct and impactful way to improve inference efficiency is to better exploit existing algorithmic sparsity through system and kernel design. **Our contributions sit exactly at this algorithm–system interface and focus on kernel‑level design and optimization.**

---

### Official Review · Reviewer_dxME · 2025-10-31

**Soundness:** 3
**Presentation:** 3
**Contribution:** 3
**Rating:** 8
**Confidence:** 4

**Summary:**

This paper presents FlashOmni, a unified sparse attention engine designed to accelerate DiTs. FlashOmni introduces flexible sparse symbols to standardize diverse sparsity strategies—including feature caching and block-sparse skipping—within a single unified framework. The paper also proposes optimized sparse GEMMs (General Matrix Multiplications) for attention computation, effectively reducing redundant operations. Experimental results show that FlashOmni maintains visual quality and achieves ∼1.5× end-to-end acceleration on the Hunyuan model.

**Strengths:**

- The paper is well-structured and clearly presents its motivation, design, and experiments.

- The topic is highly relevant, especially given the growing importance of efficient computation for large-scale video generation models.

- The idea of treating feature caching as a new dimension of sparsity is novel and elegant, providing a natural way to unify sparse attention and caching under one abstraction.

- The authors conduct comprehensive evaluations, covering both performance metrics (speedup and efficiency) and qualitative results (visual quality), which support the practicality of the proposed approach.

**Weaknesses:**

- The paper could be improved by explicitly presenting absolute end-to-end performance metrics (e.g., inference latency, total speedup) alongside visual quality metrics in a single table. This would help readers better understand the efficiency–quality tradeoff.

- (Minor) The writing and exposition could be refined to make the core idea—especially the integration of feature caching into the sparsity framework—easier to follow for readers who are not already familiar with these mechanisms and system kernel implementation.

**Questions:**

- Could the authors provide absolute end-to-end latency or throughput numbers, in addition to relative speedups?

---

> ### Author Response · Authors · 2025-11-19
>
> We thank the reviewer-dxME for the constructive comments and address the major concerns below.
>
> -----------------
> **[Weaknesses-1&Questions-1:End-to-End performance metrics]**
>
> Reply:
> Thank you for your valuable comment. Following your suggestion, we have added a new set of end-to-end performance comparisons on the Hunyuan model against existing sparse attention kernel libraries(SpargeAttn[1], DitFastAttnV2[2]) on A100. The result is shown as follows (detailed nvprof data refers to link[3]) and will be incorporated into our revised version (described in the common revision plan#1).
>
> | Method    |  Sparsity |End-to-end latency(s) | Attn latency(s)  | Preprocess latency(s) |
> |-----------|-----------|--------|--------|--------|
> | Diffusers-EfficientAttention | 0| 708  | 516.1s | 0 |
> | Diffusers-FlashAttention  |   0| 419  | 218| 0 |
> | SpargeAttn  | 32% | 401 | 171.5  | ~30 |
> | DitFastAttnV2 |  31% | 332 | 131  | 0 |
> | FlashOmni (50%,5%,6,1,30%)  | 47% | 278 | 109.1 | ~0.4|
>
> SpargeAttn is a dynamic attention kernel but suffers from insufficient overhead optimization  (dequant and judge-op in attention) and heavy pre-processing costs (dynamic mask and quant overhead for kernel) , leading to limited acceleration and reproducibility issues [4, 5]. In contrast, DitFastAttnV2 is a static sparse attention that avoids extra overhead once the kernel is selected, offering higher efficiency at equal sparsity but lower flexibility and generation quality (refer to Table 1 page 7).
>
> Our kernel also supports dynamic attention and provides **lower overhead and more efficient preprocessing compared with SparseAttn**, achieving stronger performance (more detailed analysis please refer to response  **[overhead analysis]**  to reviewer-eW6A). Moreover, it overcomes previous limitations of only supporting using block skipping or feature caching within a single layer, enabling more flexible sparse strategy combinations for improved adaptability and generation quality.
>
> -----------------
> **[Weaknesses-2:Typos and expression updating]**
>
> Reply:
> Thank you for the reminder. We will make a revision for the writing, especially part about integration of feature caching into the sparsity framework. Make the exposition and  the main contribution more clear for readers.
> (Common revision plan#4)
>
> ------------------------------
>
>
> **Reference:**
>
> [1]. Zhang, J., Xiang, C., Huang, H., Xi, H., Zhu, J., & Chen, J. SpargeAttention: Accurate and Training-free Sparse Attention Accelerating Any Model Inference. ICML25.
>
> [2]. Zhang, H., Su, R., Yuan, Z., Chen, P., Shen, M., Fan, Y., ... & Wang, Y. DiTFastAttnV2: Head-wise Attention Compression for Multi-Modality Diffusion Transformers. ICCV25.
>
> [3]./FlashOmni/example/hunyuan/nvprof/e2e
>
> [4].https://github.com/thu-ml/SpargeAttn/issues/18
>
> [5].https://github.com/thu-ml/SpargeAttn/issues/67

---

> ### Author Response · Authors · 2025-11-27
>
> Dear Reviewer dxME,
>
> We are very sorry to bother you. As the discussion period is drawing to a close (with fewer than a week remaining), we would like to ensure that we have fully addressed all of your concerns.
>
> Thank you again for your time and effort in reviewing our paper.

---

### Official Review · Reviewer_eW6A · 2025-11-01

**Soundness:** 3
**Presentation:** 3
**Contribution:** 2
**Rating:** 4
**Confidence:** 2

**Summary:**

FlashOmni presents a unified sparse attention engine designed to accelerate DiTs through combining feature caching and block-sparse skipping strategies. The paper introduces flexible 8-bit sparse symbols to standardize representation of diverse sparsity patterns, implements optimized sparse attention kernels and GEMMs, and demonstrates 1.5x end-to-end speedup on HunyuanVideo with 2-5x acceleration in individual components. The framework adopts an "Update-Dispatch" paradigm where sparse symbols are refreshed at update steps and used to guide efficient sparse computation across subsequent timesteps.

**Strengths:**

1. Unified abstraction for multi-granularity sparsity: The sparse symbol design elegantly unifies feature caching and block-sparse skipping within a single framework, addressing fragmentation in existing approaches.
2. Strong empirical performance: Achieves near-linear speedup and maintains superior visual quality compared to baselines across multiple metrics.
3. Practical kernel implementation: The general sparse attention kernel that decodes symbols at runtime eliminates the need for task-specific kernel implementations, improving engineering efficiency.

**Weaknesses:**

1. Incomplete generalization analysis: The paper claims FlashOmni is "applicable to any Diffusion Transformers," yet experiments only cover MMDiT architectures (FLUX, HunyuanVideo).

2. May lack comparison with concurrent methods: The baseline selection omits important recent work in DiT acceleration. Where is comparison with methods like DeepCache layer-wise caching variants, or other 2024-2025 methods? The related work section mentions FORA but doesn't include it in all experimental tables.

3. Insufficient analysis of quality-speed trade-offs: While Table 1 shows various configurations, there's no systematic Pareto frontier analysis. How should practitioners choose between configurations? The jump from 28% to 46% sparsity (FLUX experiments) lacks intermediate points. What is the optimal operating point for different use cases?

4. GEMM-O speedup gap inadequately addressed: The 87.5% theoretical speedup achievement for GEMM-O (vs. near-100% for GEMM-Q/attention) is mentioned but not deeply analyzed. The explanation about "multiple decoding operations along reduction axis" may be too brief or vague. Specifically: (1) Can this gap be closed with better decoding strategies? (2) What is the computational breakdown, how much overhead comes from decoding vs. irregular memory access? (3) At what sparsity level does this become the bottleneck?

5. Limited architectural diversity in experiments: All tested models share similar architectural properties (transformer-based, similar attention patterns). How does FlashOmni perform on: (1) models with vastly different sequence lengths? (2) architectures with cross-attention vs. self-attention only? (3) different token gathering strategies (the paper uses mean pooling with n=2, but what about other values or methods)?

**Questions:**

Please address the implicitly embedded questions in the weakness.

What is the trade-off curve between symbol size, decoding overhead, and sparsity granularity? Could adaptive bit-width based on sequence length or model size improve efficiency?

How did you determine the update step frequency N in your experiments?

What is the memory overhead of storing sparse symbols, cache bias, and intermediate features compared to baselines?

How robust is FlashOmni to suboptimal hyperparameter choices? Can you provide heuristics or automated methods for hyperparameter selection?

---

> ### Author Response · Authors · 2025-11-19
>
> We thank the reviewer-eW6A for the constructive comments and address the major concerns below.
>
> **[Weaknesses-1: DiT Architecture support]**
>
> Reply:
> We appreciate the reviewer’s suggestion of expanding benchmarks for DiT-based architectures. Our method focuses on optimizing the attention kernel, which **integrates feature caching directly into the attention computation, flexibly supporting all architectures**.
> The exact type of attention operation and the corresponding sparsity strategy are not fixed; instead, they are fully determined by user-defined strategies, which generate the associated sparse symbols. MMDiT is a mainstream and widely adopted DiT variant in recent work, while cross-attention has become less common, so we choose MMDiT as a case to show kernels’ performance.
>
> --------------------------------------------
> **[Weaknesses-2: Baselines]**
>
> Reply:
> We prioritized baselines from recent works (the earliest published in 2025) that demonstrate state-of-the-art acceleration performance and high adaptability to mainstream architectures.
> DeepCache[8] is closely tailored to the U-Net architecture, leveraging the property of reusing high-level features while updating low-level features at low cost. As a result, its performance for recent mainstream architectures, such as DiT-based models, is relatively limited.
>
> Regarding the feature caching method FORA[9], which is a special case with order D = 0 of TaylorSeer[10], comprehensive comparisons have already been conducted in TaylorSeer, where TaylorSeer was shown to consistently outperform FORA with the same warmup steps. Based on this, we initially included only a single comparison to avoid redundancy. Nevertheless, we greatly value the reviewer’s feedback, and in our revised manuscript, we will add further experiments with FORA and expand the corresponding Table 2 (HunyuanVideo 33K) page 7.
> Due to time constraints, we were able to complete only the first-fold evaluation at this stage. Considering that vBench metrics require multiple folds for full validation, we first performed a similarity analysis on the first fold (see Table 1, page 7). The results show that FORA performs weaker than TaylorSeer. Detailed multi-fold evaluation results will be provided in our revised version.
>  The detailed results are shown as follows:
> | Method(HunyuanVideo 33K)    |  PSNR | LPIPS   | SSIM |
> |-----------|-----------|--------|---------|
> |  FoRa (N= 6)  |  26.53  | 0.133 | 0.8596 |
> |  TaylorSeer (N= 6,D= 1)  |  26.3919   |0.1182|  0.8691|
> | FlashOmni (50%,5%,6,1,30%)   |  27.6576   |0.1185|  0.8733 |
> | FlashOmni (40%,1%,6,1,0) |  28.5233  | 0.0921 |  0.8958  |
>
> --------------------------------------------
> **[Weaknesses-3: Ablation on sparsity of FlashOmni]**
>
> Reply:
> We appreciate the reviewer’s insightful comment regarding the lack of a systematic quality–speed trade-off analysis. In response, we have added an additional set of ablation experiments, as shown in the new table. The results show that generation quality varies only marginally across different configurations, with effective sparsity exhibiting smooth and moderate changes.
>
> | Method    | Sparsity | PSNR(u)  | LPIPS (d) | SSIM(u) | FID(d) |
> |-----------|-----------|--------|---------|---------|---------|
> | ( 5%,15%, 4, 1, 0)  | 27.8%  |25.217 | 0.0999  | 0.8838 | 20.789 |
> | (20%,15%, 4, 1, 0)  |  35.7% | 25.207|  0.989 | 0.884 | 20.881 |
> | (35%,15%, 4, 1, 0)  | 39.5%  |25.178 |  0.991 | 0.8838  | 20.913 |
> | (50%,15%, 4, 1, 0)  | 41.2% | 25.159 | 0.0992 | 0.8838 | 20.933 |
>
> This can be attributed to two main factors:  (1) our sparsity schedule maintains low sparsity at the early generation stages and gradually increases it at later steps; and (2)  our sparsification strategy (Section 4.2) prioritizes updates on the most important image–text token pairs, which constitute only a small portion of all tokens; even when hyperparameter settings differ significantly, the CDF used to maintain token importance ensures that the proportion of Q blocks participating in block-skipping level computation changes smoothly. We will add these experiments into subsection 4.4 (ablation study).  (Common revision plan#3)
>
> --------------------------------------------

---

> ### Author Response · Authors · 2025-11-19
>
> **[Weaknesses-4: Performance analysis on GEMM-O]**
>
> Reply:
> We thank the reviewer for raising this important point. Compared with GEMM‑Q, where the sparsity judgement on CUDA cores is performed only once, GEMM‑Q incurs negligible extra overhead and thus can achieve nearly linear theoretical speedup. In contrast, the main bottleneck in GEMM‑O arises from decoding operations. Specifically, the sparsity judgement for GEMM‑O occurs along the reduction axis, **resulting in bitwise operations multiple times on CUDA cores**. As shown in the new table, even without sparsity, the performance gap between GEMM‑Q and GEMM‑O is already visible. The weight shape is 4096 * 3072, which is usually used in current Transformer blocks. Detailed nvprof data, please refer to link[5,6]
>
> | Seq Length(16896)  | Sparsity | Latency | Speedup |
> |-----------|-----------|--------|---------|
> | Linear  | 0 | — | 1.337 | — |
> | GEMM-Q  |  0% | 1.371 | 97.52%|
> | GEMM-Q  |  40% |  0.797| 167.8%  |
> | GEMM-Q  |  60%  |  0.525 | 254.7% |
> | GEMM-Q  |  80% |  0.268 | 498.9% |
>
>
> | Seq Length(16896)  | Sparsity | Latency | Speedup |
> |-----------|-----------|--------|---------|
> | Linear  | 0 | — | 1.377 | — |
> | GEMM-O  |  0% | 1.512 |  91.1%  |
> | GEMM-O  |  40%| 0.941  | 146.3% |
> | GEMM-O  |  60% | 0.661  | 208.3% |
> | GEMM-O  |  80% |  0.381 | 361.4% |
>
> Regarding decoding strategies, we have already applied a pre‑decoding mechanism to store masks, which effectively reduces bitwise ops times on CUDA cores. In GEMM‑O, since the bitwise ops on CUDA cores remain constant, a higher sparsity leads to a lower ratio of effective MMA computations, making the decoding overhead an increasingly significant factor in execution time.
>
> ----------------------------
> **[Weaknesses-5: Scaling analysis on Kernels]**
>
> Reply:
> We appreciate the reviewer’s insightful comment. In the revised version, we include an additional scaling analysis within the ablation experiments to evaluate the efficiency of the attention kernel across varying sequence lengths. The corresponding results are presented below.  We will add these experiments into subsection 4.4 (ablation study).  (Common revision plan#3). Detailed nvprof data please refer to link[7]
>
> | Seq length |  FlashAttentionV2(ms) |  Sparsity  |   FlashOmni-Attention| Speedup| Note |
> |-----------|-----------|--------|---------|---------| --------|
> | 2560.0  | 0.434  | 23%  |  0.356 | 1.22  | 1024 * 512 image|
> | 4608  | 1.264  | 64% |  0.604  | 2.09  | 1024 * 1024 image|
> | 9750  | 6.62 | 43% | 3.87 |  1.71 | random config |
> | 16896  | 17.986  | 58% | 8.282 | 2.17  | 2048 * 2048 image |
> | 20480  |  26.6 |  23% |  22.05 |  1.21 | random config |
> | 25152  | 42.57  | 43%  | 25.06 | 1.7  | random config |
> | 33152 |  69.616 | 64% |  27.54|  2.53 | 544 * 960 * 61 video |
> | 43890 |  128.7  | 43%  | 74.79 | 1.72 |  random config |
> | 54130 |  196.2  | 43%  | 114.8|  1.71 |  random config |
> | 66048 |276.05 |  46% | 162.32 | 1.71 | 1024 * 1024 * 61 video |
>
> Regarding the remaining two points “models involving cross-attention versus self-attention”, and “different token aggregation methods”, we provide detailed explanations in response **[DiT Architecture support]** and **[Hyper parameter selection]**.

---

> > ### Author Response · Authors · 2025-11-19
> >
> > **[Questions-1&3: Overhead analysis]**
> >
> > Reply:
> >  let H denote the number of heads, $Size_f$ the symbol size for feature caching, $Size_b$ the symbol size for block skipping, N the number of tokens (Q/K/V), L the number of transformer blocks.
> > (1). Memory overhead:
> >
> > Sparse mask:
> >
> > Our work consumes  $\mathcal{S}_s$: $  L \times (H \times \frac{N}{Size_f} \times \frac{N} {Size_b} )/ 8  $ B and $\mathcal{S}_c$: $ L \times  (H \times \frac{N}{Size_f}) / 8  $ B;
> >
> > while SpargeAttn consumes LUT:  $ L \times (H \times \frac{N}{Size_f} \times \frac{N}{Size_b}) × 4 $ B.
> >
> > And in practice, setting $Size_f$ =$ Size_b$ = 128 for the HunyunVideo model (Table 1, p.7) and generating videos with shape 544 × 960 × 61 (≈33 K tokens), our sparse symbols occupy about 11 MB, while SpargeAttn’s LUT consumes around 714 MB, **resulting in over 60× memory reduction**.
> >
> >
> > Cache Bias:
> >
> > if did’t activate the feature caching, without cost, same with block skipping methods;
> > When activating the feature caching strategy, the cache bias is used for storing the output of a transformer block(with  using GEMM-O) or attention output (without using GEMM-O), and it has the same size with feature caching-base methods.
> >
> > (2). pre-process overhead for dynamic attention kernel:
> >
> > In SpargeAttn, the per-iteration generation of the mask according to tuned thresholds, combined with QK quantization, brings a **pre-process overhead**. In this case, **this process adds 30 seconds of overhead in HunyuanVideo and 4 seconds of overhead in Flux** (see triton_bmm_pool_sim_simmean_fuse_quant item in link[1,2]). Notably, even in the Flux scenario, this additional overhead is almost close to the time cost of full attention computation itself. In contrast, in our method, the Triton kernel for generating sparse symbols and the compression kernel contribute **negligible runtime overhead** (see triton_fill_flashomni_kernel and flashomni::quantization::SegmentPackBitsKernel in the referenced link[1,2]).
> >
> >
> > (3). decoding overhead:
> >
> > We also evaluate the additional overhead within a single Attention(e.g. decoding in FlashOmni, dequant and sparsity judgement in SpargeAttn) without any sparsity.
> > Our Flux model measurements of Table 1 (p. 7)  are for the generation of *1024×1024* images (4.5K tokens). The *Giga  OPs* of $attn_{op}$ is formulated as follows, detailed nvprof data refers to reference link[3]:
> >
> > $
> > attn_{op} = 24 \times 4608 \times 4608 \times 128 \times 2 \times 2 / 10^{9} = 260.92\ \text{GOPs}
> > $
> >
> > | Method    |  Attn_Latency(ms) | TOPS   | TFOPS |
> > |-----------|-----------|--------|---------|
> > | Scale_dot     |   1.364    | 191.29 |   95.64 |
> > | FlashInfer     | 1.253    | 208.24 | 104.12   |
> > | SpargeAttn |   1.615 | 161.56 |80.78 |
> > | FlashOmni  | 1.362  |  191.57  | 95.79 |
> >
> > Our HunyuanVideo model measurements of Table 1 (p.7)  are for the generation of *544x960x61* videos (33K tokens).  The *Tera OPs* of $attn_{op}$ is formulated as follows, detailed nvprof data refers to reference link[4]:
> >
> > $
> > attn_{op} = 24 \times 32640 \times 32640 \times 128 \times 2 \times 2 / 10^{12} = 13.09\ \text{TOPs}
> > $
> >
> > | Method    |  Attn_Latency(ms) | TOPS   | TFOPS |
> > |-----------|-----------|--------|---------|
> > | Scale_dot   |   71.98   | 181.87 |   90.94 |
> > | FlashInfer   | 66.765   | 196.08 | 98.04   |
> > | SpargeAttn |   80.02 | 163.58 | 81.79 |
> > | FlashOmni  |  71.72| 182.53  | 91.27 |
> >
> > We demonstrate **superior optimization of the attention kernel** compared to SpargeAttn and **further extend support at the feature caching level**.
> >
> >
> > -----------------
> > **[Questions-2&4: Hyper parameter selection]**
> >
> > Reply:
> > In this work, our primary focus is on **innovating a high-performance fused kernel that integrates feature caching with block-sparse skipping**. The sparsity strategy and its associated hyperparameters presented in the paper serve primarily as an example to verify the quality of generation and the efficiency of kernel, even without any tuning, with all layer parameters kept fixed, our method achieves significantly better generation quality than existing baselines (refer to table 1,2 page 7). Moreover, the proposed fused kernel demonstrates substantially higher computational performance (refer to above response analysis).
> >
> > In contrast, methods such as SpargeAttn require extensive tuning and consume considerable GPU resources, as referenced in [11–14]. This highlights the robustness of **FlashOmni, which maintains superior performance  without any tuning**,  even under fixed or suboptimal hyperparameter settings.
> >
> > In future work, we plan to develop efficient hyperparameter search algorithms to automatically and quickly identify optimal configurations for further improvement in generation quality.
> >
> >
> > ------------------------------

---

> > > ### Author Response · Authors · 2025-11-19
> > >
> > > **Reference:**
> > >
> > > [1]./FlashOmni/example/flux/nvprof
> > >
> > > [2]./FlashOmni/example/hunyuan/nvprof
> > >
> > > [3]./FlashOmni/benchmark/nvprof_attn/flux
> > >
> > > [4]./FlashOmni/benchmark/nvprof_attn/hunyuan
> > >
> > > [5]./FlashOmni/benchmark/nvprof_gemmq
> > >
> > > [6]./FlashOmni/benchmark/nvprof_gemmo
> > >
> > > [7]./FlashOmni/benchmark/nvprof_attn/random_shape
> > >
> > > [8]. Ma, X., Fang, G., & Wang, X. Deepcache: Accelerating diffusion models for free.CVPR24.
> > >
> > > [9]. Selvaraju, P., Ding, T., Chen, T., Zharkov, I., & Liang, L. Fora: Fast-forward caching in diffusion transformer acceleration. arXiv preprint.
> > >
> > > [10]. Liu, J., Zou, C., Lyu, Y., Chen, J., & Zhang, L. From reusing to forecasting: Accelerating diffusion models with taylorseers.ICCV25.
> > >
> > > [11].https://github.com/thu-ml/SpargeAttn/issues/50
> > >
> > > [12].https://github.com/thu-ml/SpargeAttn/issues/6
> > >
> > > [13].https://github.com/thu-ml/SpargeAttn/issues/53
> > >
> > > [14].https://github.com/thu-ml/SpargeAttn/issues/15

---

> ### Author Response · Authors · 2025-11-27
>
> Dear Reviewer eW6A,
>
> We are very sorry to bother you. As the discussion period is drawing to a close (with fewer than a week remaining), we would like to ensure that we have fully addressed all of your concerns. If you feel your concerns have been resolved, we would greatly appreciate it if you consider raising the score.
>
> Thank you again for your time to review our paper.

---

### Author Response · Authors · 2025-11-19

**Common Revision Plan**

--------------------
We would like to thank the reviewers for their constructive comments and will make the following changes.

**[P1: End-to-End inference performance comparison]**
We will add End-to-End performance comparison to subsection 4.3 (Efficiency Evaluation)., which will include an end-to-end  inference performance comparison on HunyuanVideo(33K) A100. (shown in detailed response to reviewers-85zg & dxME).

**[P2: Overhead Analysis]**
We will add a new Overhead Analysis sub-subsection into Section 4.3 (Efficiency Evaluation). This addition will serve as a supplement to the original performance comparison, providing a more comprehensive view of efficiency. The analysis will cover three key aspects: (1) memory overhead, (2) pre-processing overhead, and (3) kernel performance (shown in detailed response to reviewers-eW6A & dxME & 85zg).


**[P3: Ablation Study]**
We will add a new Scaling Analysis and Sparsity ablation into Section 4.4(Ablation Study). (shown in detailed response to reviewer-eW6A).

**[P4: Refine the writing and exposition]**
We will make a revision for the writing, especially part about integration of feature caching into the sparsity framework. Make the exposition and  the main contribution more clear for readers.

**[P5:Typos, caption of Table and Figures]**
We will thoroughly review the confusion-related comments raised by the reviewers, improving any unclear captions accordingly and correcting those typos, especially for the TOPS metric.

**[P6: Comparing video quality with SpargeAttn with tuning-free configuration]**
We have replyed the results for the  reviewer-85zg and will add the comparison into Table-1 page 7.

**[P7: Speed comparison on more GPUs]**
We have replyed the corresponding  attention comparison results under different tokens, sparsity and headdims for the reviewer-85zg and will add the comparison into  Section 4.3 (Efficiency Evaluation).

---

### Comment · Reviewer_85zg · 2025-11-20
**1. Fix TOPS 2. Compare speed on more GPUs (RTX4090, 5090, H100) 3. Compare with tuning-free methods**

First, from the perspective of speed experiments, I have two suggestions:

1. **Correction of TOPS in the paper**

   The TOPS reported in the original version of your paper is not using the standard TOPS definition and should be fixed.

2. **Speed comparison on more GPUs**

   For a method that aims to be generally useful, it is important to demonstrate speedups on different GPUs, not only on A100. Moreover, in practice, diffusion models are often deployed on GPUs used for inference, such as RTX 4090 and RTX 5090. Therefore, the paper should also report speed comparisons on **RTX 4090**, **RTX 5090**, **H100**, etc. By the way, for SpargeAttn, you can directly benchmark using the `spas_sage2_attn_meansim_topk_cuda` API.

Additionally, some diffusion models do not use `headdim=128`, attention speed comparison on `headdim=64` is also needed.

**Lastly,** I also suggest comparing video quality with some tuning-free methods, such as the block-sparse variant of MInference and the tuning-free version of SpargeAttn, etc., for fairness.

---

> ### Author Response · Authors · 2025-11-24
>
> Thank you for your suggestion, and we have added your suggestion into our common revision plan #5, #6, #7

---

### Author Response · Authors · 2025-12-01
**Revision Change List**

We thank the reviewers for their constructive comments. Please find the revised paper and the revision cover letter below. The changes are highlighted in red.

**[R1: End-to-End inference performance comparison]**
Based on the suggestion from Reviewer-dxME and 85zg, we add a new subsection-[End-to-End Inference Comparison], page-9 into  the original Section-4.3. In our evaluation, we include comparisons with recent SOTA sparse attention kernel libraries: SparseAttn(ICML25) and DitFastAttnV2(ICCV25) with the same generation setting used in section 4.2, and report the corresponding latency with a new Table 3 page-9.

**[R2: Memory Overhead]**
Based on the suggestion from Reviewer-eW6A, we add a new subsection [Memory Overhead] page-9 for overhead analysis of sparse symbols and cache bias into original Section-4.3 [EFFICIENCY EVALUATION]. The pre-processing overhead is included in [R1] and the kernel performance is included in [R7]


**[R3: Ablation Study]**
Based on the suggestion from Reviewer-eW6A, we provide the ablation studies about  Sparsity ablation into  the original Section-4.4, page 10 [ABLATION STUDY], with updating Table 4 page-10.

**[R4: Refine the writing and exposition]**
Based on Reviewer-dxME’s suggestion, we have made revisions to clarify and strengthen the core idea of FlashOmni. In particular, we have refined the explanation of how feature caching is integrated into the sparsity framework, and we now more clearly articulate both the motivation for this design and its potential to enable further optimization of sparse GEMM operations. The detailed revisions include: Section-1, page 1,2 & Section-3.4, page 5 & Section 3.5, page 6.

**[R5: Address reviewers’ confusions]**
Based on the suggestion from Reviewer-85zg, we made the following clarification.

1. In Section-4.1, page-7, we clarify the definition of the TOPS metric, which is used in SpargeAttn, and correct the corresponding data in Table-1, page 7. (Reviewer-85zg)

2. We revise the caption of Figure-5 & 7 & 10 & 11, which is about the kernel performance (From Inference performance to  kernel latency). (Reviewer-85zg)


**[R6:More comparison on video generation quality]**
Based on the suggestion from Reviewer-eW6A and 85zg, we have updated the original Table-1, page 7 & Table-2, page 8 about the comparison on Video Generation (SpargeAttn untuned:topk=73% and FORA:N=6)

**[R7: Kernel Performance]**
Based on the suggestion from Reviewer-eW6A and 85zg, we add the following experiment.

1. Scaling Analysis with different sequence length: We add a new paragraph into  the original Appendix-4, page 16, to include kernel performance across varying sequence lengths, with a new Table 7, page 16. (Reviewer-eW6A)

2. Speed comparison on 4090 and different headdim:  We add a new paragraph into  the original Appendix-4, page 16, to include kernel performance comparison on 4090 with different headdim, with a new Table 8, page 16. (Reviewer-85zg)

3. Performance analysis on GEMM-O: We add a new paragraph into  the original Appendix-5, page 17, to include performance analysis for speedup gap between GEMM-Q/O, with a new Table 9, page 17. (Reviewer-eW6A)

---

### Author Response · Authors · 2025-12-02
**Summary Comments**

We sincerely thank the Area Chair for the time and effort invested in our submission. Here is a brief summary.

----------

Reviewers **eW6A** & **dxME** mainly requested additional experiments and clearer writing. In our rebuttal, we conducted the key experiments they proposed and refined the presentation accordingly.

Reviewers **pugu** and **85zg** initially misunderstood FlashOmni’s novelty and requested more experiments, our rebuttal and discussion addressed their concerns, provided the requested results, and led both to raise their scores.

Below, we provide a concise reviewer-by-reviewer summary of the main concerns and how we addressed them, which we hope will assist the AC in evaluating our revision. **The detailed revision info please refer to the revision change list below**.

--------------------

About **Reviewer-eW6A**

Reviewer eW6A’s concerns focus on two aspects:

**1.Questions regarding the small details**

I. About support scope: FlashOmni focuses on a high-performance sparse kernel that flexibly supports all architectures, including those suggested by the reviewer. [Weaknesses-1]

II. About choice of hyperparameters: Hyperparameter tuning is not the main focus; even with fixed (sub-optimal) untuned hyperparameters, FlashOmni still outperforms existing baselines. [Questions-2&4]

**2. Hoping for additional Ablations and performance analysis.**

We provide corresponding experiments in our rebuttal and add these into our revision pdf, including:

I. Revision#6 (R6) for more baseline (Weaknesses-2: W-2);

II. R3 for ablation on sparsity (W-3);

III. R2 and R7.3 for the overhead analysis on attention and performance analysis on GEMM‑O (W-4 and Questions-1&3);

IV. R7.1 for the efficiency of the attention kernel across varying sequence lengths (W-5).

More details  please refer to revision change list and revised paper.

These experiments directly address the reviewers’ concerns and show that FlashOmni achieves state-of-the-art kernel efficiency while maintaining strong generation quality. We believe we have adequately addressed all the concerns of reviewer-eW6A in our rebuttal.


--------------------

About **Reviewer-dxME**

Reviewer dxME’s concerns focus on two aspects:

**1. Hoping end-to-end experiments**

we provide corresponding experiments in our rebuttal and add these into our revision pdf, including:

I. Revision#1 for the end-to-end performance comparisons (Weaknesses-1 & Question-1).

**2. Suggestion about refining writing**

We are very grateful for this suggestion. We make the revision about the part of core design to make the exposition and the main contributions clearer for readers (Weaknesses-2). Detailed revision please refer to Revision#4.

We believe we have adequately addressed all the concerns of reviewer-dxME in our rebuttal.

--------------------

About **Reviewer-pugu**

Reviewer pugu’s concerns focus on two aspects:

**1.Misunderstanding the core contribution of FlashOmni.**

Reviewer Pugu’s original comments may suggest that limited familiarity with the infrastructure track (our submission track) or with kernel‑level optimization (the main focus of FlashOmni), and tends to view work in this area as “simple engineering” (Weaknesses-1,2,3).
In response, we clarified key point and highlighted our main contribution:

I. A unified sparse attention that treats feature caching as a new dimension of sparsity into the sparsity framework to support multiple sparsity granularities within a single kernel, while existing sparse attention only supports block-sparse skipping.

II. Sparse GEMM-Q/-O leverages this feature caching sparsity to unlock previously untapped potential for eliminating redundant computations, a capability absent from existing sparse kernels.

III. Leveraging optimizations in both Attention and GEMM, FlashOmni achieves end-to-end high-quality generation alongside high-performance inference by supporting flexible, multi-granularity sparsity designs.

We have made the explanation, supported by extensive data, in our rebuttal and addressed the concerns. In revision#4, we also make revisions for the content that previously created ambiguity and make the exposition and the main contributions clearer for readers.

**2. Not familiar with similarity metrics.**

Reviewer pugu questioned the absence of PSNR, LPIPS, etc. for the full‑attention baseline, but these metrics are typically reported only for sparse methods relative to the baseline and thus omitted for the baseline itself. We clarified this misunderstanding to him, and he understood and agreed with our explanation.[Question-1]


Reviewer-pugu stated that his concerns had been addressed and subsequently increased his score.

--------------------

---

> ### Author Response · Authors · 2025-12-02
> **Summary Comments#2**
>
> About **Reviewer-85zg**
>
> Reviewer 85zg’s concerns focus on three aspects:
>
> **1.Misunderstanding the core contribution of FlashOmni.**
>
> We explained the motivation of FlashOmni, the effectiveness of the proposed sparse symbols in reducing memory consumption (Revision#2), and the implementation challenges and efficiency benefits of sparse Gemm [Weekness1,2,3]. And the main contribution we explained in our rebuttal is similar to the summary of **Reviewer-pugu** #1.
>
> In Revision #4, we also make revisions for the content that previously created ambiguity and make the exposition and the main contributions clearer for readers.
>
>
> **2. Hoping for more comparative experiments with SpargeAttn.**
>
> We have provided the detailed data in our rebuttal and made the corresponding revision in pdf:
>
> I. Revision#1(R1) for the end-to-end speedup experiments [W-4]
>
> II. R7.2 for the speed comparisons on more GPUs with different headdim.  [Suggestion-2]
>
> III. R6 for the video generation quality against untuned SpargeAttn. [Suggestion-3]
>
> The experimental results show that FlashOmni consistently outperforms both tuned and untuned SpargeAttn under various settings.
>
> **3. Suggesting to use the same TOPS metric as SpargeAttn**
>
> In revision#5, we have clarified the mismatch of the definition for TOPS, supported by extensive data in our rebuttal, and corrected it in the revised version.[Questions-1]. As for the performance (250TOPS) of SpargeAttn without any sparsity on A100, mentioned by reviewer[Questions-1], we provide the nvprof data and prove that the data is unable to reproduce; there are also GitHub issues in the SpargeAttn repository raising similar concerns.
>
>
> Reviewer-85zg ultimately increased his score directly without leaving any comments.

---

### Meta-Review · Area_Chair_xQ4E · 2025-12-24

**Summary:**

The paper proposes a technique to combine feature caching and sparse attention to further speed up diffusion transformers. The contribution includes a framework to do so and optimized kernel implementations based on FlashInfer, which shows better memory and compute efficiency than previous sparse attention techniques such as SpargeAttn.

While I believe the contribution and improvements in this paper is valid. There are some concerns remain (see reviewer concerns). I make the decision mostly based on Concerns 1 and 3. Currently, the manuscript still looks much like an implementation trick, putting the sparse bit compression (32 bit mask -> 1 bit mask) at the main position. While I believe the paper has some good merits (both algorithmic and kernel side) beyond trick, the manuscript cannot be accepted in the current form.

**Reviewer Concerns:**

1. the novelty is limited on the sparse and feature caching algorithm, mostly on the implementation side (pugu, 85zg)
The authors also acknowledge that their contribution is mostly on the kernel side, including mostly storage efficiency and improvements on GEMM-O / GEMM-Q kernel implementations.

2. requires more diverse architectural evaluations (eW6A)
The authors explains why they only evaluate on the MMDiT architecture.

3. writing (eW6A, dxME, 85zg)
The authors revised the manuscript significantly during rebuttal. However, I think even with the revisions, reviewer's concern remains on the core contribution of the paper is not clear. Furthermore, the main novelties, including the unified sparse attention-feature caching framework, should be more elaborated. I think the bulk of improvement of this works comes from combining SA and FC. However, the challenge of doing so, the technical shining spot, and more detailed performance decomposition should be included in the writing.

4. possibly unfair/nonisolated comparison (pugu, 85zg)
The authors provide many experiments. I believe their superiority is valid. However, to what extent the performance gain comes from better kernel, and to what extent the performance gain comes from combining FC and SA, is still not clear to me.

**Reviewer Scores:**

I think except dxME has an original rating of 8. Other 3 reviewers would at most give a rating of 4 after rebuttal. Leaving an averaging rating of 5.

---

### Decision · Program_Chairs · 2026-01-26

Reject